# The Potential of *Ginkgo biloba* as a Source of Biologically Active Compounds—A Review of the Recent Literature and Patents

**DOI:** 10.3390/molecules28103993

**Published:** 2023-05-09

**Authors:** Patrycja Biernacka, Iwona Adamska, Katarzyna Felisiak

**Affiliations:** Faculty of Food Science and Fisheries, Department of Food Science and Technology—West Pomeranian University of Technology, 70-310 Szczecin, Poland

**Keywords:** ginkgo biloba, patents, ginkgotoxin, pro-health properties, food industry

## Abstract

*Ginkgo biloba* is a relict tree species showing high resistance to adverse biotic and abiotic environmental factors. Its fruits and leaves have high medicinal value due to the presence of flavonoids, terpene trilactones and phenolic compounds. However, ginkgo seeds contain toxic and allergenic alkylphenols. The publication revises the latest research results (mainly from 2018–2022) regarding the chemical composition of extracts obtained from this plant and provides information on the use of extracts or their selected ingredients in medicine and food production. A very important section of the publication is the part in which the results of the review of patents concerning the use of *Ginkgo biloba* and its selected ingredients in food production are presented. Despite the constantly growing number of studies on its toxicity and interactions with synthetic drugs, its health-promoting properties are the reason for the interest of scientists and motivation to create new food products.

## 1. Introduction

The fossil remains of plants of the Ginkgoaceae family are well known to paleobotanists: representatives of this family lived 300 million years ago (in the Permian period), and they achieved the greatest importance in the Jurassic period (200 million years ago). Currently, only *Ginkgo biloba* L. (Figure 1) is a naturally occurring species in this group. This plant survived the mass extinctions in the Cretaceous and Paleogene periods and the Pleistocene glaciation, becoming a relict and endemic species in China [1].

Due to its high ornamental and medicinal value, ginkgo has been spread all over the world. It was favored by enormous adaptability to the environment, high resistance to air pollution and almost all pests and pathogens. The high viability of this species due to the duplication of genes responsible for resistance and stress reactions made it ideal for use in urban greenery arrangements. It is now commonly planted around the world in university campuses, parks and gardens, or along streets and sidewalks [1,2]. These trees are also a source of artistic and religious inspiration for the inhabitants of many continents [1]. Old specimens are commonly found in temples, old villages, or near streams in East Asia. Ginkgo is considered in China as a cultural symbol of hope and peace and is called the national tree of China [2].

*Ginkgo biloba* leaves are a popular herbal medicine registered in the Chinese Pharmacopeia (2015 edition) [3]. The preparations made from them are used, inter alia, in the treatment of cardiovascular and cerebrovascular diseases. The effectiveness in alleviating cardiovascular ailments was confirmed in the 1960s [4].

*Ginkgo biloba* leaf extract is one of the best-selling herbal remedies in the world and the most-sold herbal supplement in the US and Europe. It has great therapeutic potential, including scavenging free radicals, reducing oxidative stress, as well as reducing damage to the nervous system and reducing platelet aggregation. It also has anti-inflammatory, anti-cancer and anti-aging properties. Clinical studies also confirm its beneficial effects, among others, in the treatment of central nervous system disorders—Alzheimer’s disease and cognitive deficits [5].

The aim of the publication is to analyze the results of research mainly from 2015–2022 on the chemical composition of *Ginkgo biloba* and its biological activities, as well as toxicity and interactions with drugs. This publication contains a section devoted to the analysis of granted patents in the field of innovative possibilities of using Ginkgo biloba in the production of food and beverages.

## 2. Phytoconstituents of Plant

*Ginkgo biloba* contains many compounds with a unique structure that can be used in herbal medicine. These include, for example, terpene trilactones (ginkgolides), acylated flavonol glycosides (ginkgoghrelins), biflavones (ginkgetin), ginkgotides and ginkgolic acids [6]. 

Ginkgo extract contains over 60 bioactive ingredients, but the most important role is played by flavonoids and terpenoids. They usually constitute about 24% and 6% of the extract, respectively. Moreover, it contains organic acids, proanthocyanidins, tannins, sitosterols, carotenoids, polysaccharides, glucose and other ingredients (minerals and vitamins) [7].

### 2.1. Terpenoids

Plant secondary metabolites are a group of small-molecule organic compounds produced as a result of the secondary metabolic activity of plants. These substances are stored in certain organs or tissues of plants, are species-specific and are involved in the stress resistance of plants and the transmission of information. The main terpenoids present in ginkgo are bilobalides (sesquiterpene) and ginkgolides (diterpenes), which are the only ones to contain t-butyl [C_17_(CH_3_)_3_]. They are natural substances with functional groups that play an important role in the protection and treatment of cardiovascular and cerebrovascular diseases. Bilobalides and ginkgolides are present in all parts of the ginkgo seeds they contain, and the highest total terpenoid content was found in the embryo and endosperm [8].

So far, ten diterpenoid lactones have been discovered, named ginkgolides and labeled Q, P, N, M, L, K, J, C, B and A. The group of sesquiterpene lactones is bilobalide, and its isomers contain two lactone ring groups. In addition to these two groups of substances, *Ginkgo biloba* also contains nor-terpenoids, including three nor-sesquiterpenoids [9,10,11]. The terpenoid fraction of the extract consists mainly of ginkgolides A, B, C, J and M (about 2.8–3.4%) and bilobalides (2.6–3.2%) [12].

Ginkgolides have a high medicinal value. The terpene trilactones present in ginkgo, including ginkgolides A, B, C and bilobalide, correspond, inter alia, to its anti-epileptic effect on neurons within the brain’s hippocampus, improving memory and learning ability and ameliorating neuronal damage. Especially important is ginkgolide B. This terpenoid shows high biological activity due to its role as an antagonist of the platelet-activating factor receptor. This compound has antioxidant, anti-inflammatory and anti-apoptotic effects [13]. DeFeudis et al. (2003) [14] found that ginkgo extract, and especially ginkgolide B, inhibited the proliferation of the very aggressive human breast cancer cell line and xenografts of this cell line in mice. However, the inclusion of ginkgolide B in therapy may be associated with mild side effects, including headache, somnolence, hiccups, and general weakness. Ginkgolide C, on the other hand, has a different effect: its application may contribute to the reduction of lipid storage [7].

It is also important that the flavonoid and terpenoid fractions of ginkgo extracts can act in a complementary manner, inhibiting several processes related to carcinogenesis in the development of neoplastic diseases [14].

### 2.2. Flavonoids

Flavonoids are important natural bioactive compounds with a strong influence on the human body. *Ginkgo biloba* leaves contain a number of substances from this group, including flavonol glycosides, biflavones, proanthocyanidins and isoflavonoids. However, the majority are the multiform glycosides of quercetin, kaempferol, and isorhamnetin. Flavonoids are the main constituents of ginkgo leaf extract [8].

One hundred ten flavonoids belonging to seven classes have been identified in ginkgo extracts. The first class consists of 52 glycosides of flavonols and seven flavonols. Known aglycones of flavonol glycosides include quercetin, kaempferol and isorhamnetin. In addition, from the group of aglycons, there are also syringetin, myricetin, laricitrin, myricetin 3′,4′-dimethyl ether and patuletin. The second class consists of 14 flavone glycosides and five flavones. The third class included two flavanones and one flavanone glycoside, the fourth class—two isoflavones and one isoflavone glycoside, and the fifth class—four flavan-3-ole. The sixth class consisted of 13 biflavonoids, and the seventh consisted of nine biginkgosides [7].

Ginkgo flavonoids and their glycosides exhibit multidirectional biological activity, including antioxidant, anti-cancer, anti-bacterial, anti-viral, anti-inflammatory and neuroprotective properties. A strong therapeutic effect was shown by the combination of phenolic aglycones of quercetin, kaempferol or isorhamnetin [7,15].

*Ginkgo biloba* seedlings, up to 5 years old, contain more flavonoids and terpenoids than adult trees. Therefore, the young leaves are used to produce a standardized extract (EGb761). With the age of trees, the content of biologically active ingredients decreases, and, thus, the quality of extracts produced from them. In recent years, there has also been interest in increasing the content of flavonoids in *Ginkgo biloba* leaves, e.g., through foliar fertilization or alternative partial irrigation of the root zone [16].

### 2.3. Carboxylic Acids

Organic acids are common chemical components of plants that are characterized by high biological activity. Preparations made of *Ginkgo biloba* contain about 13% carboxylic acids, including quinic acid, chlorogenic acid, ascorbic acid, shikimic acid, gallic acid, protocatechuic acid, vanillic acid, isovanillic acid, coffee acid, sinapinic acid, ferulic acid, 6-hydroxybenzoic acid, p-coumaricic acid and p-hydroxybenzoic acid [7,11,17,18,19]. In addition, phenolic acids in *Ginkgo biloba* leaves also occur in glycosidic or covalently bonded forms [19]. In ginkgo leaves, quinic acid is the most occupied—2.26 g/100 g of dry weight. Shikimic acid is also present in large amounts—2.24 g/100 g dw. Malic acid was the least—0.58 g/100 g dw [20].

The organic acids present in *Ginkgo biloba* have a very strong free radical scavenging effect. Flavones and procyanidins are also characterized by the same activity [21]. Studies have shown that protocatechuic acid present in ginkgo has the ability to induce terminal kinase-dependent hepatocellular carcinoma cell death and increase the endogenous antioxidant potential of macrophages, and gallic acid exhibits antitumor activity [7].

### 2.4. Lignins

The richest part of ginkgo in lignin is the shells surrounding the seeds. The content of these substances can be as high as 40%. At the moment, 24 lignans and their isomers have been isolated from this plant [22].

Although lignins have interesting physicochemical properties and high biological activity, they are not often used due to the secondary metabolite of lignocellulosic biomass. Lignin consists mainly of three phenylpropane units: p-hydroxyphenyl (H), guaiacyl (G) and syringyl (S). It increases the strength and rigidity of lignocellulose cell walls and provides a physical barrier against phytopathogen invasion and other environmental stresses. This means that lignin can be considered a bioactive macromolecule [11,22].

The lignins isolated from *Ginkgo biloba* include i.e. sesamin, ginkgool, pinoresinol, ginkgolide B, and lariciresinol. Their action is mainly based on antioxidant activity [9,10,11].

### 2.5. Proanthocyanidins

Proanthocyanidins are highly active, functional polyphenolic compounds. They are oligomers or polymers of a polyhydroxy flavan-3-alcohol [e.g., (+)—catechins and (-)—epicatechins)] and flavan-3,4-alcohol linked by a single C4-C8 or C4-C6 bond (type B) or by an additional C2-O-C7 or C2-O- bond C5 (type A) [23].

Proanthocyanidins constitute 4–12% of ginkgo leaves, and standardized extracts contain 7% of proanthocyanidins. Although studies on the composition of these compounds are still ongoing, it has already been shown that proanthocyanidins and flavan-3-oils have antioxidant activity and the ability to scavenge free radicals. Moreover, they alleviate ischemic-reperfusion damage conditions and exhibit antihypertensive, anti-atherosclerotic and anti-aggregating, immunomodulating, antiseptic and anti-inflammatory effects [17].

### 2.6. Polyprenols

Polyprenols consist of 12–20 cis- and two trans-isoprene units and one form of betulaprenol and are terminated in an isoprene unit (having a hydroxyl group). Polyprenols mainly occur as a mixture of homologs in the photosynthetic organs of plants and have a similar structure and composition to dolichols [24]. *Ginkgo biloba* leaf polyprenols are weakly polar unsaturated polyisoprenoid alcohols found in leaf lipids. There they occur mainly in the form of polyprene acetate [25]. Ginkgo leaf polyprenols exhibit antioxidant, immunomodulating, anti-bacterial, anti-viral, antitumor and hepatoprotective properties [24,25,26].

### 2.7. Polysaccharides

Among the many bioactive compounds found in ginkgo, there are also polysaccharides. Purified polysaccharides are obtained from ginkgo by extraction methods (hot water extraction, ultrasound-assisted and enzymatic extraction) and by purification methods (including ion exchange chromatography and gel filtration). Large amounts of structurally diverse polysaccharides, mainly in terms of monosaccharide composition, were isolated from both leaves, sarcotesta and seeds. However, most of them consist of rhamnose (Rha), galactose (Gal), mannose (Man), xylose (Xyl), arabinose (Ara), glucose (Glu) and fucose (Fuc) with different mole fractions of the individual components [11,27]. Ginkgo seeds are a rich source of mannose (Man), and the sarcotesta contains more galactose (Ga) and glucose (Glu) compared to leaves. Interestingly, the molecular weight of *Ginkgo biloba* polysaccharides shows a varied distribution ranging from 1.0 kDa to 5679 kDa [27].

Polysaccharides isolated from ginkgo have antioxidant properties, as well as anti-cancer, anti-inflammatory, hepatoprotective, antidepressant, immunostimulating and even anti-alopecia properties [28,29].

### 2.8. Alkylphenols and Alkylphenolic Acids

Alkylphenols occurring in the leaves of *Ginkgo biloba* can be divided into five groups: cardanols, α-hydroxycardanols, cardans, urushiols and isourushiols, and a group of alkylphenolic acids, which include ginkgolic acids. These compounds are among the toxic ingredients of *Ginkgo biloba*. Ginkgolic acid occupies a special place here, as it is considered to be toxic, mutagenic and sensitizing. However, despite their negative effect, a beneficial pharmacological effect on the human body was also shown for example, ginkgolic acid C17:1 in studies showed various antitumor effects [6].

### 2.9. Other

Sixty-eight chemical compounds have been identified in the composition of essential oils obtained from *Ginkgo biloba* leaves [30], among which the largest percentage was sesquiterpenes (42.11%) [7].

Ginkgo seeds and leaves are a source of vitamins B, C and E. Among the minerals, a relatively high content of zinc, iron, sodium, magnesium, potassium, calcium, carbon and nitrogen was found. According to research by Pereira et al. (2013) [20], among the macronutrients, carbohydrates (72.98 g/100 g dw) have the largest share in *Ginkgo biloba* leaves. The protein content is 12.27 g/100 g dw, the ash content is 12.27 g/100 g dw, and the fats have the smallest share—4.75 g/100 g dw. The content of free sugars such as fructose, glucose and sucrose was 1.42; 0.78; 0.23 g/100 g dw, respectively, a total of 2.43 g/100 g dw free sugars.

The approximate calorific value of *Ginkgo biloba* dried leaves is 287 kcal/g [31]. In turn, according to the study by Tomowa et al. (2021) [32], the protein content of *Ginkgo biloba* seeds was 5 g/100 g of raw seeds or about 11 g/100 g dw. The fat content in raw nuts was 1 g/100 g, and in dry matter, 2.04 g/100 g. The content of saturated acids was 19 g/100 g, polyunsaturated 40 g/100 g, and monounsaturated 41 g/100 g. Starch was isolated as seed after extraction with organic solvents and water. The average yield of the product after extraction was about 70 g/100 g dw of *Ginkgo biloba* seeds. Information on selected biologically active substances is presented in Table 1.

## 3. Structure and Biosynthesis of Ginkgolides and Bilobalides

Ginkgolides have a similar molecular formula. Some have the notation C_20_H_24_O (A, B, C, M, J, P, Q, N), and some C_20_H_22_O (K, L) (Table 2) [34,40,43,44]. These compounds have a rare group in natural products. These are six 5-membered rings, including a spiro [4,4] carbocyclic nonane ring, three lactones, a tetrahydrofuran ring, and a tert-butyl moiety. All ginkgolides have a similar structure and differ in substituents R1, R2 and R3, which are permutations of H or OH [114].

Bilobalide was first isolated by a Major in 1967 [115]. It is characterized by a tendency to isomerization under mild acylation conditions to form diacyl derivatives of the spiro compound. Bilobalide, also belonging to the diterpenoids, also have a tert-butyl group.

Although the health-promoting properties of the terpenoids found in ginkgo have been thoroughly investigated, their biosynthesis is still not fully understood. One of the enzymes most often involved in ginkgolides biosynthesis is Gb LPS (levopimaradiene synthase), a diterpene synthase that catalyzes the synthesis of levopimaradiene (2)—possibly the precursor to all ginkgolides. Ginkgolides and bilobalides share a common three-step biosynthetic pathway. The first step involves the biosynthesis of two simple five-carbon units that build the isoprene skeleton of isopentyl diphosphate (IPP) and its isomer dimethylallyl diphosphate (DMAPP). In the second step, there is a repetitive condensation of IPP and DMAPP towards farnesyl precursors (FPP) and geranylgeranyl diphosphate (GGPP). This is followed by late cyclization and oxidation steps catalyzed by terpenoid synthases and cytochrome P450 (CYP-450) dependent monooxygenases, which define the specific carbon backbone and oxidation pattern of the product [114].

The cytosolic pathway of mevalonic acid in the cytosol, from 3-acetyl-CoA to IPP, is responsible for the synthesis of sesquiterpenoids and sterols, while for the formation of monoterpenoids, diterpenoids components—the plastid pathway of methylerythritol 4-phosphate producing IPP and dimethylallyl diphosphate from pyruvaldehyde-glyceraldehyde-3-glycerate [116]. This section may be divided by subheadings. It should provide a concise and precise description of the experimental results, their interpretation, as well as the experimental conclusions that can be drawn.

**Table 2 molecules-28-03993-t002:** Research on the biological activity of *Ginkgo biloba* (2015–2022).

Type of Activity	Substance	Result	Source
anti-inflammatory	Ginkgolides	reducing nerve inflammation and slowing down the progression of the disease (Guillain-Barré syndrome) reduction of Th17 cells reduction of interferon (IFN)-c and interleukin-12 (IL)-12 levels	[54]
	Extract	Reversal of reduction of synaptopodine and nephrine Activation of heme oxygenase-1 (HO-1) Reduction of TNF-α, IL-6 and fibronectin enhancement Reduction of lipid accumulation in the kidneys Reduction of ROS production in cells	[55]
	Ginkgolides	regulation (blocking) of protein kinase signaling pathways by reducing the activity of MAPK (mitogens) and NF-κB (nuclear factor kappa B). reducing signal transduction by blocking the PAF mediator (platelet activating factor)	[53]
	Leaf extract (EGb 761)	Inhibition of the expression of the pro-inflammatory cytokines IL-1, IL-6 and TNF-α	[51]
	Bilobalide	Reduction of the level of inflammatory cytokines: interleukin 6 (IL-6) and IL-1β Reduction of the mRNA expression level in colon Weakening of the kappa B (NF-kB) nuclear factor signaling pathway Reduction of tumor necrosis factor (TNF-a) in serum	[52]
	Ethanol extract of flowers bilobetin isoginkgetin	Lowering the level of nitric oxide (NO) and increasing the NO inhibition ratios Reduction of prostaglandin E2 (PGE2) and interleukin-6 (IL-6) levels Reduction of TNF-α, iNOS mRNA and COX-2 mRNA levels	[50]
	Ginkgolide B	Reduction of the expression of inflammatory cytokines in RAW264.7 macrophages Reduction of the expression of NOX4, monocyte chemotactic protein-1 (MCP-1), intercellular adhesion molecules-1 (ICAM-1) and vascular cell adhesion molecules-1 (VCAM-1) in cells	[48]
	Amentoflavone	Regulation of p38 pathways, NF-κB and Jun1 and Jun2 N-terminal protein kinase Reduction of inflammation in cells infected with serotype 2 (SS2)	[57]
	GBSP3a (water-soluble polysaccharide)	Reduction of expression of pro-inflammatory mediators and cytokines in macrophages RAW264.7 [nitric oxide (NO), prostaglandin E2 (PGE2), tumor necrosis factor-α (TNF-α), interleukin-6 (IL-6), interleukin-1β (IL-1β)] Modulation of the NF-κB and MAPK signaling pathways (suppression)	[104]
	Ginkgolide A	reduction of the nuclear factor kappa-B (NF-κB) and mitogen-activated protein kinases (MAPK) activation of the AMPK protein kinase inhibition of the release of TNF-α and IL-6 inhibition of the expression of pro-inflammatory mediators (COX-2 and NO) lowering the levels of pro-inflammatory cytokines TNF-α, IL-6 and IL-1β in macrophages and monocytes	[47]
	Extract EGb 761	improvement of the general condition of the body in chronic colitis weight increase improvement of colon parameters (length and weight) inhibition of pathological changes in the large intestine reduction in the number of cariocytes reduction of the expression of IL-7, IL-6, TNF-α and IFN-γ (inflammatory factor proteins) and mRNA	[10]
	Ginkgolide A	Inhibition of TLR4-NF-κB signaling through PI3K/AKT pathway Reduction of release of inflammatory mediators and activation of NF-κB signaling Reduction of TLR4 mRNA expression without reducing cell viability	[46]
	Leaf extract (IGbE-761^®^)	Inhibition of NO and PGE2 production Inhibition in the mRNA and protein expression levels of inducible iNOS and COX-2 enzymes Lowering the level of pro-inflammatory cytokines (IL-1β, IL-6 and TNF-ɑ) in macrophages Inhibition of NF-kB activation	[45]
	Ginkgo biloba leaf polysaccharides (PGBL)	Inhibition of the translocation of nuclear factor (NF)-kB into the nucleus of cells Reduction of the expression of tumor necrosis factor (TNF- ɑ) and interleukin-6 (IL-6) at the protein and mRNA levels	[44]
	Ginkgolide A	Inhibition the endothelial production of high-glucose-induced interleukin (IL)-4, IL-6, IL-13 and signal transducer and activator of transcription-3 (STAT-3) phosphorylation	[46]
anti-bacterial	Leaf extract (GLE)	microorganisms groups tested: *Akkermansia, Alistipes, Alloprevotella, Anaerotruncus, Bacterioides, Bacteroidetes, Blautia, Colidextribacter, Dubosiella, Erysipelatoclostridium, Faecalibaculum, Firmicutes, Lachnoclostridium, Lachnospiraceae* UCG-006, *Lachnospiraceae* NK4A136 group, *Parabacteroides, Rikenellaceae* RC9 gut group, *Roseburia,* norank *Desulfovibrionaceae*, norank *Lachnospiraceae,* norank *Muribaculaceae*, unclassified *Lachnospiraceae* change in the composition of the intestinal microbiota change in *Firmicutes/Bacteroidetes* ratio (decrease) increase in the number of bacteria from *the Akkermansia, Alloprevotella, Alistipes* and *Parabacteroides* groups	[73]
	Leaf extract (GLE)	microorganisms tested: gut microbiota in mice significantly affects the amount and composition of intestinal bacteria increase in the diversity of the intestinal microflora reduction of the number of bacteria belonging to the groups *Proteobacteria* and *Deferribacteres*	[63]
	Water extract Chloroform extract Methanol extract	microorganisms tested: *Bacillus subtilis, Enterobacter aerogenes* ATCC 13048, *Enterococcus durans*, *Escherichia coli* ATCC 259222, *Klebsiella pnemoniae, Listeria innocua*, *L. monocytogenes, Salmonella enteritidis* ATCC 13075, *S. infantis, S. typhimurium, Staphylococcus aureus* ATCC 25923, *St. epidermidis* DSMZ 20044 methanol and chloroform extracts inhibit the activity of all tested microorganisms water extracts did not affect the activity of *S. entiritidis, S. infantis, L. innocua* and *L. monocytogenes* minimal inhibitory activity against *E. aerogenes, S. infantis, S. aureus, S. epidermidis, B. subtilis, E. coli* was demonstrated at a concentration of 50 mg/mL of extract the lowest concentration inhibiting the development of 99.9% of bacterial strains is 100 mg/mL	[62]
	Ethanol extract of leaves	microorganisms tested: *Bacillus thuringiensis* CCM 19, *Clostridium perfringens* CCM 4991, *Escherichia coli* CCM 3988*, Haemophilus influenzae* CCM 4456, *Klebsiella pneumoniae* CCM 2318, Listeria monocytogenes CCM 4699, Salmonella enterica subsp. *enterica* CCM 3807*, Shigella sonei* CCM 1373, *Staphylococcus aureus* subsp. *aureus* CCM 2461*, Yersinia enterocolitica* CCM 5671 anti-bacterial activity against all tested bacteria the highest effectiveness against *S. aureus, E. coli, K. pneumoinae* and *Y. enterocolitica*	[64]
	Gelatin film with the addition of ginkgo extract (GBE)	microorganisms tested: microorganisms tested: *Staphylococcus aureus* ATCC 6538, *Candida albicans* ATCC 10231 inhibition of the activity of the tested microorganisms	[67]
	Ginkgetin	microorganism tested: *Streptococcus suis* direct binding to suilysin (cytolysin produced by *S. suis*), which prevents protein oligomerization, reduces hemolytic activity and protects cells from damage	[117]
	Ginkgetin	reducing the amount of pro-inflammatory cytokines and modification of TLR4/NF-κB signaling reduction of inflammation caused by cerebral ischemia Reduction of the expression of cyclooxygenase-2 (COX-2) and induced nitric oxide synthase (iNOS) Reduction of the expression of interleukins IL-1β, IL-6 and IL-8, tumor necrosis factor alpha (TNF-α) and prostaglandin E2 (PGE2) Increased expression of interleukin-10 (IL-10)	[58]
	Leaf extracts (GLE)	microorganisms tested: *Saprophytic staphylococcus, Shewanella putrefaciens* reduction of the activity of *S. putrefaciens and S. staphylococcus* destruction of the structure of bacterial cells; cells with damaged membranes and cell walls create aggregations; cells die more damage to *S. putrefaciens* cell membranes than *S. staphylococcus* minimal inhibitory concentration for both microorganisms was 100 mg/mL, inhibition rates were higher for *S. putrefaciens* as compared to *S. staphylococcus*	[66]
	Amentoflavone	microorganism tested: *Streptococcus suis* reduction of *S. suis*-induced cytotoxicity in macrophages reduction of the number of tested bacteria in the organisms of mice lowering mouse mortality	[57]
	Ginkgolic acid (GA) C15:1 monomer	microorganisms tested: *Bacillus amyloliquefaciens* SQR9 CGMCC 5808*, Escherichia coli* DH5α ATCC53338, *E. coli* O157: H7 ATCC43895, *Pseudomonas aeruginosa* PAO1 ATCC15692*, P. putida* KT2440 ATTC47054, *Ralstonia solanacearum* ATCC11696, *Rhodococcus jostii* RHA1, *Streptococcus thermophilus* ND03, *S. aureus* ATCC25923 no effect on the activity of Gram-negative bacteria inhibition of the activity of gram-positive bacteria inhibition of the biosynthesis of DNA, RNA and proteins in bacteria cells	[60]
	Polyprenol (GBP)	microorganisms tested: *Escherichia coli* NCTC 12923, *Staphylococcus aureus* ATCC 25923 Increased anti-bacterial activity Extension of the anti-bacterial time Increased anti-bacterial activity of antibiotics	[61]
Antioxidant	supernatant obtained after mixing the fermented seed powder and saline	fermentation increased the antioxidant activity Initial decrease in antioxidant activity (2nd day of fermentation), then a marked increase (maximum value—4th day of the process), and then another decrease Reduction of the content of harmful ginkgolic acids (decrease by 45%)	[94]
	hydroethanolic leaf extract and ingredients: flavone, ginkgolide, procyanidins, and organic acids	DPPH scavenging ability was highest with procyanidins and lowest with ginkgolide. Flavone ability was lower than procyanidins and higher than organic acids. scavenging capacity of ABTS was the highest in the case of flavone and the lowest in the case of ginkgolide, while procyanidins—lower than that of flavone and higher than that of organic acids antioxidation shows synergistic effects the highest scavenging of ABTS and DPPH radicals was obtained in a solution of flavone: procyanidins in the proportion of 1: 9	[21]
	Leaf extract (EGb 761)	EGb 761 exhibits antioxidant activity enhances the action of drugs used in diseases of the nervous system	[74]
	Ethanol extracts	antioxidant activity was determined by DPPH and MRP methods; in addition, TPC and TFC have been demonstrated extracts contain a number of ingredients with antioxidant properties all extracts showed antioxidant activity differences were shown in the antioxidant activity prepared from green and yellow leaves (yellow leaves were more active)	[64]
	Leaf extract (EGb 761)	ginkgolides A-C, kaemferol, quercetin, bilobalide and isorhamnetin determine the high antioxidant activity of the extract regulation of the expression of antioxidant enzymes (increased synthesis) reduction of the amount of ROS and RNS which lowers lipid peroxidation	[51]
	Ethanol extract	antioxidant properties were determined by DPPH and ABTS methods; inhibitory effect on MMP-1 and reactive oxygen species was determined extracts contain many ingredients with an antioxidant effect (e.g., quercetin, kaempferol and ginkgolides A-C) extracts showed antioxidant activity (DPPH: 0.103 mg/mL; ABTS: 0.052 mg/mL)	[73]
	extract (GBE)	antioxidant properties were determined by DPPH method the addition of GBE increased the antioxidant activity (the scavenging effect increased from 24.7%	[67]
	Polysaccharides GBPS-2 and GBPS-3	low antioxidant capacity determined by the method of scavenging hydroxyl radicals and DPPH (GBPS-2 and GBPS-3) at high scavenging capacity of superoxide radicals and ABTS (GBPS-2 and GBPS-3)	[72]
	ginkgo biloba (10 mg/kg/day)	the levels of MDA and GSH in the aortic tissues were determined ginkgo biloba exhibit antioxidant activity, as demonstrated by lowering the level of MDA and increasing the level of GSH in the aorta of animals on a high-cholesterol diet	[73]
	methanol extract from leaves ethanol (40%, 70% and 96% *v*/*v*) extracts from leaves	antioxidant activity was determined by the DPPH method leaves at the end of vegetation (yellow) had a high free radical scavenging capacity the extracts prepared with 40% and 70% ethanol and methanol extracts had higher antioxidant capacity than the extracts prepared with 96% ethanol the method of extracting plants (using a Soxhlet apparatus or rotary shaker) significantly influences the antioxidant activity of the extracts	[70]
	polysaccharide monomers	the antioxidant activity of solutions containing polysaccharide monomers was tested (GBP, GBP’, GBP11, GBP22, and GBP33) by DPPH, ABTS and superoxide anion methods GBP22, GBP and GBP11 have strong antioxidant effects potential ingredient of cosmetics and functional foods	[56]
	extract EGb 761	Expression levels of NADPH oxidases (NOXs), NADPH oxidase activity, oxidative stress through the levels of glutathione (GSH), malondialdehyde (MDA), nitric oxide (NO) and superoxide dismutase (SOD) in brain tissues were determined EGb-761 shows high antioxidant activity there are possibilities of using this substance of natural origin in medicine (treating diseases of the nervous system)	[69]
Anti-cancer	bilobol isolated from fruit	cytotoxic to 293, B16F10, BJAB, and HCT116 tumor cells (in vitro; doses 15.0–50 μg/mL) increased expression of caspase-3 and caspase-8 in HCT116 (human colon cancer cells) and tumor cell apoptosis	[87]
	Ginkgetin (extract)	marked reduction in the viability of HepG2 and SK-HEP-1 cells inhibition of the cell cycle of cancer cells (HepG2 and SK-HEP-1) in S phase tumor cell apoptosis (growth of apoptotic bodies and reduction of tumor cell size; HepG2 and SK-HEP-1) increased caspase-3 activity and cytochrome c release, without increased caspase-8 activity inhibition of tumor growth	[83]
	Methanol extract from kernel	decreased viability (cytotoxic effect) of HCT116 and A2058 tumor cell lines no cytotoxic effect on McCoy-Plovdiv non-cancer cells (cell proliferation at all tested extract concentrations: 0–800 μg/mL)	[82]
	Ginkgo biloba extract (EGb-761)	gastric cancer cell (GC SGC-7901 and MGC-803) proliferation was inhibited reduced migration and invasiveness of cancer cells lowering the level of MMP2, p-ERK1/2, NF-kB p-P65 and NF-kB P65 proteins in neoplastic cells stunted tumor growth inhibited metastasis of gastric cancer to the liver	[81]
	Substances isolated from fresh male flowers: Amentoflavone 7″-O-β-d-glucopyranoside Amentoflavone Bilobetin Isoginkgetin Sciadopitysin	Anti-proliferative activity against the HeLa, HepG2, NCI-H460, Daudi, K562, SKOV3, MIAPaca-2, MCF-7 tumor cell lines was examined bilobetin and isogingetin had the strongest anti-proliferative effects against different tumor cell lines HeLa cells were the most sensitive to bilobetin and isogingetin (induction of cell apoptosis) inhibition of the tumor cell cycle in the G2/M phase activating the protein Bax and executive caspase-3 bilobetin inhibited the production of Bcl-2 (a protein with an anti-apoptotic effect)	[86]
	Ginkgolide B	decrease in the maximum inhibitory concentration (IC50) of gemcitabine in tumor cells of the BxPC-3, CAPAN1, PANC1 and MIA PaCa-2 lines in combination with gemcitabine: inhibiting cell proliferation; inhibiting tumor growth; increasing cell apoptosis; no effect when gemcitabine and extract are used separately suppression of the effect of gemcitabine in the following areas: increase in NF-kB activity and PAFR and phosphorylated NF-kB/p65 expression suppression of PAFR expression no effect on the IC50 of gemcitabine in IκBα-SR	[86]
	Leaf extract IDN 5933	there were no adverse clinical effects leading to liver or thyroid tumors liver injury markers remained constant there was no difference in the expression of the c-myb, p53, and ctnnb1 genes	[80]
	Amentoflavone	strong reduction in the synthesis of tumor necrosis factor alpha (TNF-), interleukin-1 (IL-1) and IL-6 in cells infected with Streptococcus suis	[57]
	Methanolic extract from leaves	hepatocellular carcinoma (HCC) activity chemically induced in rats (N-nitrosodiethylamine; in vivo research) increase in the level of ING-3 in liver cells decreased expression of the Foxp-1 gene in the liver reduction of serum AFP, CEA and GPC-3 levels positive histological changes in the liver tissue	[79]
	Polysaccharide isolated from leaves (Se-GBLP)	decrease in the viability of human bladder cancer cells T24 inducing apoptosis of cancer cells by reducing the expression of the anti-apoptotic protein Bcl-2, increasing the expression of the pro-apoptotic protein Bax, inactivating caspase-3, caspase 9 and PARP, reducing the activity of mitochondria (limiting the activity of mitochondrial membranes)	[13]
	Extract EGb 761	reduction of aromatase activity in MCF-7 cells with overexpression of aromatase reduction of cytochrome p450 aromatase (CYP19) mRNA and decrease in protein expression (especially the CYP19 I.3 and PII promoter) reduction of 17β-estradiol levels in MCF-7 AROM cells reducing the size of the tumor reduction of CYP19 mRNA expression in the tumor	[78]
	Ginkgolide B (GB)	suppressing the invasion of bladder cancer neoplastic cells by reducing the expression of the ZEB1 protein as a result of increasing the level of miR-223–3p	[85]
	Extract EGb 761	inhibition of the proliferation of gastric cancer cells AGS (stopping development in the G0/G1 phase) inducing cell death; increase in the proportion of dead cells (apoptosis) to living cells lowering the level of Bcl-2 increasing the expression of caspase3 and p53	[76]
	Extract EGb 761	inhibited expression of KSR1, p-KSR1, ERK1/2 and p-ERK1/2 enhancing the inhibition of tumor cell proliferation increasing the induction of tumor cell apoptosis reduction of malondialdehyde (MDA) levels in cancer cells increased activity of superoxide dismutase (SOD) and glutathione peroxidase (GSH-Px) in cancer cells	[77]
	Ginkgolide B (GKB)	alleviating inflammation in the colon reduction in the number and size of the tumor decrease expression of vascular endothelial growth factor (VEGF) decrease in micro vessel density (MVD) in the tumor	[84]
	Extract EGb 761	strong induction of apoptosis in human melanoma cells as a result of imbalance between pro and anti-apoptotic proteins from the Bcl-2 family no apoptotic changes in melanocytes caspase and mitochondrial pathway-dependent apoptosis (caused by lower mitochondrial membrane potential and greater activation of Bak and Bax) lowering the level of Mcl-1 in melanoma cells resulting in stronger apoptosis of cancer cells	[35]
Anti-obesity, anti-atherogenic and anti-diabetic	Leave extract (GbE)	alleviating hypercholesterolemia, inflammation and atherosclerosis caused by following a high-fat diet change in the quantitative proportions of microorganisms in the intestinal microflora reduction of intestinal transcription of pro-inflammatory cytokines; reduction of intestinal inflammation, improves the state of the intestinal barrier promoting the production of short-chain fatty acids, indole-3-acetate and secondary bile acids, the presence of which is associated with areas of atherosclerotic plaque the transplant of the altered intestinal microflora defeated atherosclerosis	[65]
	Extract GbE	beneficial changes in body composition (amount of body fat) positive effect on the level of adiponectin positive effect on the blood lipid profile reducing the anxiety index and increasing latency to immobility	[90]
	Extract (GbE)	beneficial changes in kidney function associated with reduction of renal glomerular hypertrophy; reduction of the kidney/body weight ratio and reduction of albuminuria inhibition of the reduction of synaptopodine and nephrine increase in HO-1 expression in the kidney reduction of the accumulation of TNF-α, IL-6, fibronectin and lipids in the renal glomeruli reducing the uptake of low-density oxidized lipoprotein and reducing the production of ROS in podocytes with high glucose levels	[55]
	Ginkgo biloba seeds (GBS)	regulation of glucose and lipid metabolism lowering fasting blood glucose and serum insulin levels improvement of glucose and insulin tolerance antioxidant and anti-inflammatory effect	[91]
	vinegar obtained from fermented coats of ginkgo seeds	a reduction in weight gain associated with a high-fat diet reducing the size of fat cells inhibited differentiation of adipocytes by: reduced expression of proteins involved in adipogenesis: C/EBPδ and PPARγ, inhibition of lipid accumulation in 3T3-L1 cells induced to convert to adipocytes	[97]
	Extract GbE	reducing energy consumption reduction in the size of the adipocytes reduction of acetate accumulation reduction of [3H]-oleate incorporation into adipose tissue	[96]
	Extract GbE	lowering the level of HbA1c in the blood lowering of fasting serum glucose and insulin decreased body mass index (BMI), waist circumference and visceral adiposity index	[89]
	Ginkgo biloba leaves	increasing the reduction of triglycerides, total cholesterol, and low-density lipoprotein (LDL-C) cholesterol; increase in the level of high-density lipoprotein cholesterol during the currently used therapies with statins, simvastatin and atorvastatin	[93]
	Ginkgolide B	reduction of LOX-1 expression in umbilical vein endothelial cells (HUVEC) and RAW246.7 macrophages; reducing cholesterol deposits in them reduction of NOX4 expression inhibition of expression of intercellular adhesion molecule-1 (ICAM-1), vascular cell adhesion molecule-1 (VCAM-1) and monocyte chemotactic protein-1 (MCP-1) in umbilical vein endothelial cells reduction of inflammatory cytokine expression in RAW264.7 macrophages at the transcriptional and protein levels decreased expression of matrix metalloproteinase-1 and cyclooxygenase-2 in RAW264.7 macrophages	[48]
	Ginkgolide C	reduction of lipid overaccumulation in HepG2 cells intensification of triglyceride breakdown as a result of increased lipase expression of fatty triglycerides and increased lipase phosphorylation reduction of fatty acid synthesis in hepatocytes as a result of stimulation of CPT-1 to activate b-oxidation of fatty acids, increase of sirt1 and phosphorylation of kinase and reduction of acetyl-CoA carboxylase expression	[95]
	Ginkgo biloba leaves	improvement of the parameters of the lipid profile lowering the levels of hsCRP and ICAM-1 in the serum as inflammatory markers induced by a high cholesterol diet lowering the level of MDA in the aortic tissue increase in GSH levels in the aortic tissue reduction of atherosclerotic changes	[71]
	Ginkgolide B	effect on metabolic disorders in mice with obesity induced by a high-fat diet activation of hPXR transactivity decrease in body weight and serum triglyceride (TG) levels reduction of fatty liver (improvement of lipid accumulation) increase in mRNA expression of target PXR genes in the liver	[74]
	Extract GbE	reduction of food consumption (as a source of energy) reduction of weight gain increase in the expression of Adipo R1 and IL-10 genes increase in IR and Akt phosphorylation reduction of NF-𝜅B p65 phosphorylation and TNF-𝛼 levels	[88]
Neuroprotective and anti-neurodegenerative	Extract EGb	beneficial effect in Alzheimer’s patients, especially women decreased production of TNF-α, IFN-γ and IL-10 increase in the production of IL-15 and IL-1β lowering the expression of SOCS1, SOCS3, IRF-3, IRF-7, tetherin, NFKB1, p65 and MxA genes	[106]
	Extract EGb 761	alleviation of cognitive impairment occurring in mild and moderate forms of Alzheimer’s disease	[103]
	Ginkgo biloba dropping pill (GBDP) Extract EGb 761	preventing the loss of dopaminergic neurons in Parkinson’s disease improvement of cognitive functions related to neuronal damage activation of the Akt/GSK3b pathway (neuroprotective effect) GBDP pills were more effective in treating Parkinson’s than EGb 761 extract	[104]
	Extract EGb 761	with simultaneous administration with donepezil: increased pro-cholinergic and antioxidant effect, which leads to a marked improvement in memory (cognitive functions) no changes in plasma or uptake by brain of donepezil or bilobalide	[74]
	Extracts (GB, EGb 761) Tablets	beneficial effect on the functioning of memory, including cognitive functions in neurodegenerative diseases (Alzheimer’s and Parkinson’s) and in cancer improved blood supply to the brain, which improves the ability to concentrate beneficial effect on executive functions and non-verbal memory soothing mood changes and reducing stress	[105]
	Extract EGb 761	treatment of neurodegenerative diseases by reducing cell apoptosis as a result of: ○ decrease in acetylation of p53 lysine 382, ○ increase in the potential of the mitochondrial membrane, ○ reduction of the BAX/Bcl-2 ratio reduction of cleavage of PARP [Poly (ADP-ribose) polymerase]	[102]
	Ginkgolides	disruption of the signaling pathways of the nuclear factor NF-κB and MAPK kinase responsible for inflammation occurring in neurological diseases reduction of inflammation as a result of inhibition of signal transduction by altering the activity of the PAF activating factor	[53]
	Extract EGb	improvement of cognitive memory function in mild symptoms of Alzheimer’s dementia	[3]
	Extract EGb 761	reduction of inflammatory processes in primary microglial cells occurring in Alzheimer’s disease by: ○ inhibition of prostaglandin E2 release ○ reduction of mPGES-1 protein synthesis ○ change in the levels of proinflammatory cytokines ○ decrease in cytosolic activity of phospholipase A2	[101]
	Extract EGb 761	neuroprotective and antioxidant effect in neurodegenerative diseases Alzheimer’s and Parkinson’s	[98,100]
	Extract EGb 761	effective and safe in the treatment of dementia and Alzheimer’s disease in a daily dose of 240 mg	[99]
Protection of sense organs	GBE capsule (120 mg: 27% flavone glycosides + 6.8% terpene lactones from ginkgo)	vascular protection: favorable morphological changes of the vessels in radial peripapillary capillary of the eyes (increased density)	[109]
	Ginkgo leaf tablets	effective control of the rate of retinopathy and disease progression (type 2 diabetes) improvement in the rate of remission (extension of the asymptomatic period)	[107]
	Extract EGb 761	prevention of hearing impairment when taking cisplatin during chemotherapy (confirmed by comparison of the amplitude of the distortion products of otoacoustic emissions and the signal-to-noise ratio)	[108]
Cardiovascular protection	Ginkgolide B	decreased expression of cyclooxygenase-2 and etalloproteinase-1 in RAW264.7 macrophages (contributes to the protection against atherosclerosis)	[48]

## 4. Pharmacological Activities

*Ginkgo biloba* has been used for years as a herbal plant supporting memory processes. Initially, it was studied in terms of neuroprotective and anti-neurodegenerative effects. However, as a result of the analysis of the composition of the extracts obtained from the leaves and the influence of these extracts or their selected components, it was found that this plant has a wide multidirectional effect on the functioning of the organism. As a result of studies conducted on animals and on human tissue lines, it has been shown that ginkgo extracts and tablets and their selected ingredients exhibit anti-inflammatory, anti-bacterial, antioxidant, anti-cancer, anti-obesity, anti-diabetic, anti-atherogenic, cardioprotective and oto-protective effects (Table 2).

### 4.1. Anti-Inflammatory Effect

Extracts prepared in laboratories and their commercial formulas, such as individual ingredients: ginkgolides (A or B), bilobalide, amentoflavone, and water-soluble polysaccharides, were tested for anti-inflammatory properties. All studies have shown a positive damping effect on the developing inflammation. The most commonly observed reductions in nitric oxide, interferon, prostaglandin E2, TNF-α, IL-1, IL-4, IL-6, IL-12, and IL-1β were observed in inflamed tissues [118,119,120,121,122,123,124,125,126,127,128,129,130], as well as inter alia, changes in MAPK and NF-κB signaling pathways [116,122], caused, inter alia, by weaker translocation of the nuclear factor NF-κB [131,132]. In addition, there is also increased activation of AMPK protein kinase [122] and heme oxygenase [130].

### 4.2. Anti-Microbial Activity

The use of ginkgo for anti-bacterial purposes has been the subject of research for a long time. Initially, the effectiveness of extracts prepared from various parts of plants was analyzed—e.g., fruit, leaves and roots, as well as selected components of these extracts, e.g., ginkgo acids or free phenolic acids for the few taxa of bacteria. The most frequently tested microorganisms were *Escherichia coli* and *Staphylococcus aureus.* In these studies, inhibition of the activity of selected bacterial taxa was demonstrated, and the results became the basis for further research on anti-bacterial activity. In recent years, ginkgetin [129,133], amentofavone [132], ginkgolic acid C15:1 monomer [134] and polyprenol [135] and the effectiveness of leaf extracts obtained with the use of various solvents (water, ethanol, chloroform and methanol) [136,137,138,139]. The group of microorganisms has been significantly expanded, including taxa of gram-positive and gram-negative bacteria [125,126,134,136,138,140,141], intestinal microflora typical of the tested mammalian organisms [137,139], as well as human pathogenic fungi (e.g., *Candida albicans* used in the study by) [141]. In all the studies carried out, different effects on the activity of microorganisms were shown, and the strength of the effect depends on the tested pathogen and the dose/amount of the substance used. Moreover, higher efficiency of alcoholic extracts than water extracts was found [136].

### 4.3. Antioxidant Activity

The study of antioxidant activity was carried out on the supernatant obtained from fermented ginkgo seeds, leaf extracts obtained with the use of various solvents, as well as their individual components (including polysaccharides and their monomers, organic acids, procyanidins, flavone and ginkgolide). These analyzes were carried out using various methods (e.g., DPPH, ABTS, scavenging hydroxyl radicals or superoxide anion methods), which does not allow for a direct comparison of the results of all these studies. However, all studies showed the antioxidant activity of the tested substrates, often assessed as high or very high [21,126,131,138,141,142,143,144,145,146,147,148,149]. However, it was noted that this activity varies depending on the date of leaf harvest: it is the highest in the case of raw material harvested in autumn [68]. Both commercial EGb761 extracts [143], as well as methanol and ethanol extracts prepared under laboratory conditions, are highly active. In the latter case, extracts made in alcohol at a concentration of 40% and 70% are more effective compared to extracts made in the presence of alcohol with a concentration of 96% [144]. Among the tested extract components, procyanidins and flavones contained in ginkgo leaves showed the highest antioxidant activity [21].

### 4.4. Antitumor Activity

The research on antitumor activity was carried out using the tissue culture method, which consisted of treating selected tumor cell lines (Table 2) with selected substances. When analyzing the results of studies published mainly in 2015–2022, it can be concluded that work on the use of ginkgo extracts or their selected ingredients is widely conducted, but it should be intensified. In total, the effect of the extract and seven selected substances found in ginkgo was analyzed and tested on at least 22 cancer cell lines; however, usually, the effect of the selected substance on only 2–3 tumor cell lines was analyzed. Nevertheless, the discussed studies demonstrated cytotoxicity or inhibition of selected development phases of neoplastic cells that inhibit their proliferation, both when using extracts obtained from leaves [42,150,151,152,153,154,155,156,157], as well as selected components isolated from them: ginkgolide B [158,159,160] and polysaccharides isolated from leaves [13], bilobol isolated from the fruit [161] and chemical components isolated from fresh male flowers, incl. amentoflavone and its derivatives [134,160], bilobetin, isoginkgetin and sciadopitysin [160]. Very often, the effect of substance application was the improvement of parameters indicating the stage of disease development, which resulted in a slowdown or complete inhibition of tumor development (Table 2).

### 4.5. Anti-Obesity, Anti-Atherogenic and Anti-Diabetic

During the research, a positive, antiatherogenic, anti-obesity and anti-diabetic effect of ginkgo on the metabolism of mammalian organisms was observed. The effects of both leaf extracts [130,139,162,163,164] and seeds [165], vinegar obtained from fermenting seeds [166], leaves [148,167], well as ginkgolide B [123,168] and ginkgolide C [169] were investigated. During the research, positive changes in body weight (decrease) were observed [163,164,170], and a reduction in adipose tissue mass [167,168,169] was caused by the reduction of fat cells (adipocytes) [169,170]. Blood tests showed positive changes in the fat profile [145,164], including lowering cholesterol and triglycerides [167,168], and the regulation of lipid metabolism, glucose and insulin [166,167] had a positive effect on the condition and functioning of the kidneys [130]. Additionally, a study by Wang et al. (2022) [139] showed that the oral administration of *Ginkgo biloba* leaf extract is effective in relieving hypercholesterolemia, systemic inflammation and atherosclerosis, and these effects are associated with the modulation of the taxonomic composition of intestinal microbes, protection of the integrity of the intestinal mucosa, and improvement of microbial metabolic phenotypes (Figure 2).

### 4.6. Neuroprotective and Anti-Neurodegenerative

In studies on the influence of ginkgo on the functioning of the nervous system, mainly ginkgo leaf extracts were used [3,140,171,172,173,174,175,176,177,178,179,180,181,182], less often tablets [176,177]. The use of all these types of preparations was conducive to the reduction of inflammatory processes within the nervous system [128,170], the number of nerve cells subject to damage and apoptosis [169,173] and the improvement of blood circulation within the cerebral vessels [171]. The therapies resulted in the improvement of memory, especially cognitive memory [3,168,169,170,171,172], which was of great importance for limiting and inhibiting neurodegenerative changes, especially in Alzheimer’s and Parkinson’s diseases. It is also important that these preparations enhance the effect of a drug traditionally used in the treatment of dementia symptoms occurring in neurodegenerative diseases—donepezil [140].

### 4.7. Protection of Other Organs

Research has shown that both EGb extract [179] and ginkgo leaf tablets [180,181] have a protective effect on the sensory organs. This is of great importance, especially for patients suffering from type 2 diabetes, because it allows to delay or completely inhibit damage to blood vessels in the eye’s retina leading to visual disturbances [179], as well as contributing to at least partial regeneration and improvement of the condition of these vessels [181]. In addition, the substances contained in the ginkgo leaf extract protect the hearing organ against damage caused by taking cisplatin during chemotherapy implemented during the treatment of neoplastic diseases [180]. One of the components found in ginkgo leaves, ginkgolide B, also has a protective effect on blood vessels, preventing the occurrence of atherosclerotic lesions [123].

## 5. Toxicity

Toxic components in *Ginkgo biloba* include alkylphenols. Their classification is presented in Section 5 (Phytoconstituents of the plant). These compounds are a mixture of several 2-hydroxy-6-alkylbenzoic acids. One of the toxic components is ginkgolic acid, designated as C13:0, C15:1 and C17:1 [182].

*Ginkgo biloba* standardized extract EGb761 is classified as a therapeutic agent for the treatment of the central system, the main one in the treatment of dementia, but also helpful in the treatment of Alzheimer’s and Parkinson’s. It is credited with relieving symptoms, memory functions and handling, dizziness, migraines, or tinnitus. To be able to attribute the pro-health effect of EGb761, it should contain 22–27% of flavonoids and 5–7% of terpenoids and less than 5 ppm ginkgo biloba acid (i.e., 0.0005% ginkgo biloba acid in the preparation) [6,32].

The seeds of this plant are used in Asian cuisine for the production of stuffing, soups, desserts, meat and vegetarian dishes, and the roasted seeds are a popular delicacy. While eating cooked ginkgo seeds is safer than eating them raw, they can be toxic if consumed in large amounts or over a long period of time, especially in children. Ginkgo seeds contain a toxic component of MPN (4-methoxypyridoxin) called ginkgotoxin, and in addition, ginkgo seed tests contain large amounts of alkylphenol (over 4% ginkgolic acid)—eating more than 10–20 nuts a day may pose a health risk [32]. However, it has been shown that the concentration of ginkgotoxin in the protein of ginkgo seeds increases during the growing season and reaches its maximum in early August, but then its content drops sharply. Canned and cooked seeds now contain only about 1% of ginkgotoxin present in raw seeds, which can be attributed to their water solubility. On the other hand, the content in roasted seeds is slightly lower than in raw seeds because the compound is thermally stable [6,183]. For this reason, when using *Ginkgo biloba* extract preparations, it is important to ensure the safety of patients.

According to the research of Gawron-Gzella et al. (2012) [184], the pharmacopoeial requirements for ginkgo leaf extract refer to the number of bilobalides and the sum of A, B and C ginkgolides, while the manufacturers of preparations indicate the content of total terpene lactones on the labels. Research shows that many manufacturers do not always keep the declared total of terpene lactones (6%), and the preparations do not always contain the correct portion of bilobalides and the sum of ginkgolides. The content of ginkgolic acid, with the applicable norm below 5 ppm, in dietary supplements was very often overstated, sometimes even 1600 times. As a result of consuming such a large amount of ginkgolic acid, problems with the digestive system (nausea, vomiting, diarrhea), headaches and dizziness, palpitations, anxiety, weakness or skin allergy. In the case of people with blood clotting problems and/or taking non-steroidal anti-inflammatory drugs, antiplatelet or anticoagulant medications, it can lead to internal hemorrhage [185]. Ginkgotoxin poisoning can also occur. The content of ginkgotoxin in seeds ranges from 170 to 404 μg/g. Concentrations above 170 μg/g cause toxicity and manifest as seizures, loss of consciousness and leg paralysis [186]. It is important that such poisoning can be prevented with vitamin B6: administration of a dose of 30 mg of pyridoxal 5′-phosphate (corresponding to 2 mg/kg of body weight) causes the symptoms of poisoning to cease [187].

The results of Boeteng & Yang (2021) [24] showed that the number of toxic compounds in fresh Ginkgo biloba seeds (ginkgotoxin, ginkgolic acid and cyanide) was significantly reduced during seed drying. The ginkgotoxin content was reduced by a factor of four, and the amounts of acrylamide, ginkgolic acid and cyanide in the dried seeds were reduced to a safe level (safety range). Of the four drying methods tested, radiant drying turned out to be the most effective: it lasted the shortest, and the obtained product showed the highest quality and content of bioactive compounds, as well as the strongest antioxidant activity.

Recently, the attention of scientists has been attracted by the possibility of using alkylphenols for medical purposes, which in appropriate doses, have beneficial effects, including anti-cancer and anti-bacterial properties [134].

A study by Borenstein et al. (2020) [187] has been shown to inhibit Herpes simplex type 1 virus multiplication, human cytomegalovirus genome replication and Zika virus infection. In addition, it inhibits the synthesis of all three classes of HIV, Ebola, Influenza A, and Epstein–Barr virus fusion proteins. The results also indicate that inhibition of virion entry by blocking the initial fusion event following ginkgolic acid administration post-infection suggests a possible secondary mechanism targeting protein and DNA synthesis. This is confirmed by the strong action of this acid, effective even after the infection process has taken place. The results also indicate the possibility of using it in the treatment of acute infections (e.g., caused by coronavirus, Ebola virus, Zika, influenza A and measles), as well as active local lesions (e.g., caused by HSV-1, HSV-2 and varicella viruses—VZV shingles).

The publication of Omidkhoda et al. (2019) [188] discusses the protective effect of using *Ginkgo biloba* leaf extract in case of poisoning caused by various factors: natural toxins (scorpion venom, lipopolysaccharides, aflatoxin B1, lysophosphatidylcholine, pentacyclic triterpenoids, cassava, cotton seed pigment called gossypol), chemical toxins (metals): aluminum, lead, cadmium, mercury; heavy metals contained in aqueous waste, fluorine, triethyltin, ethanol, carbon tetrachloride, pesticides, chemotherapeutic drugs, cigarette smoke, naphthalene or monosodium glutamate) and radiation. The beneficial effect of the extract on the poisoned organism is probably related to the high antioxidant activity of the extract (manifested by the reduction of lipid peroxidation and restoration of reduced dehydrogenases, glutathione peroxidase, superoxide dismutase and catalysis) and its anti-inflammatory effect.

According to the results of in vitro studies, biflavonoids (ginkgetin, isogingetin, amentoflavone, sciadopitysin and bilobetin) can also be toxic to the body. They were observed to be cytotoxic to human proximal tubular cells and to be less toxic to healthy human liver cells. In addition, activated apoptosis was associated with biflavonoid-induced nephrotoxicity. These data suggest that these biflavonoids exhibit potential hepatic and renal toxicity [24].

The supplement market should be more regulated so as not to lead to accidental poisoning. In addition, the production should be strictly regulated to ensure that such a supplement will not contain nutrients or be contaminated. Products, such as infusions, are a safe products as a kind of supplementation with a balanced diet. Despite the risks, special care should be taken by pregnant women and children.

## 6. Interactions of *Ginkgo biloba* Extracts with Drugs

*Ginkgo biloba*, as a raw material belonging to phytotherapeutic drugs, shows mainly antioxidant and neuroprotective properties. It contains pharmacologically active ingredients that may be useful in treating many diseases, but due to its antiplatelet effects, it may interact with other antiplatelet drugs (warfarin, aspirin) or herbal preparations with similar antiplatelet effects (garlic or ginseng) [189]. According to Kedzia and Alkiewicz (2006) [190], ginkgo preparations in combination with aspirin may cause hematomas in the anterior chamber of the eye and with paracetamol—subdural hematomas.

Research by Bogacz et al. (2016) [189] proved that extracts from this plant could modulate the expression of cytochrome P450 enzymes and, thus, influence transcription factors, thanks to which they can participate in the metabolism of xenobiotics (drugs, procarcinogens, vitamins and food components).

Ginkgo preparations may also accelerate the metabolism of omeprazole and esomeprazole, primarily by influencing the mechanism of CYP2C19 induction and consequently reducing the effectiveness of these drugs in preventing upper gastrointestinal bleeding. In addition, they increase the risk of bleeding while taking SSRIs or SNRIs [191]. Single cases of coma in humans have been shown to be caused by the concomitant intake of *Ginkgo biloba* preparations with trazodone, and cases of priapism have been observed as a result of an interaction between *Ginkgo biloba* and risperidone. It was also noted that the use of *Ginkgo biloba* may reduce the concentration and effectiveness of valproate and reduce the anxiolytic and hypnotic effects of benzodiazepines [192,193]. Woroń & Siwek (2018) [194] proved that the combination of ginkgo biloba with dormitive and/or anxiolytics or with fluoxetine caused side effects in the form of dizziness, somnolence and hypotension.

In the case of non-standardized extracts prepared in accordance with the European Pharmacopoeia, the effect of the extract remains uncertain. A significant drug interaction potential cannot be ruled out in the case of poorly standardized ginkgo leaf extracts used in many dietary supplements. A review of research to date shows that *Ginkgo biloba* extracts are very reactive. Therefore, patients should be checked for health prior to administration, and any possible signs of drug interactions should be carefully considered [6].

## 7. Patents

*Ginkgo biloba* has been used in Chinese medicine for centuries, but recently, the leaves and fruits of this plant have become objects of interest in the pharmaceutical, food and cosmetic industries. The greatest industrial interest in Ginkgo biloba leaves and fruits occurred at the end of the last century when several hundred patents were issued, mainly in Japan, China and the USA. However, at the beginning of the 21st century, more than a thousand patents were published each year, with the largest number of publications appearing in 2017 (over 7000); in the following years, the number decreased, with 391 patents published in 2021 and 275 in 2022 (state on 7 November 2022). Generally, referring to the Espacenet patent database (European Database Espacenet), during the last three decades, more than 29 thousand patents appeared all over the world. Most of them concern the application of ginkgo extracts in medicine, methods of extraction or preparation of tablets or pills.

Until 1990 of the 44 patents listed for the entry “*Ginkgo biloba*” in the Espacenet patent database (European Database Espacenet), many patents were related to cultivation and chemicals, as well as drugs, for example, anti-vomiting preparations or anti-inflammatory medicines, which contain ginkgo extract.

In the years 1990–2000, the content increases in the number of patents (458 patents) concerning mainly the extraction methods of valuable substances from *Ginkgo biloba* leaves [195,196,197,198,199,200,201] and the use of the extracts in medicine [179,180,181,182,183,184,185,186,187,188,189,190,191,192,193,194,195,196,197,198,199,200,201,202]. There have also been patented food products containing ginkgo biloba leaves, like tea mixtures [203] or drinks [204] and other products enriched with leaves extract, e.g., chewing gum [205], chocolate [206], and candies [207].

The new inventions concerned mainly extraction methods allowing to obtain extracts with a reduced content of toxic compounds, such as 4′-o-methyl pyridoxine and biflavones, alkyl polyphenols or ginkgolic acid. Extracts were usually obtained from ginkgo leaves using organic solvents, e.g., acetone or methanol, which were removed in the next steps of processing, and the resulting concentrate was dissolved in water, ethanol or other solvents, with the steps of purification and filtration to remove alkyl polyphenols [208]. The patented production process of extract EGb 761^®^ involves extraction of a mixture of leaves from China, France, and the USA with 60% (m/m) aqueous acetone, acetone removal by evaporation, cooling with stirring the aqueous solution to precipitate chlorophylls, biflavones and most of the ginkgolic acid [23,202,209]. Extracts prepared using organic solvents are treated with lead compound (e.g., lead acetate) or insoluble polyamide than the filtrated solution is extracted with an aliphatic solvent, the aqueous-alcoholic solution is concentrated, ammonium sulfate is added, the solution is extracted with methylethylketone and ethanol. The concentrated, filtrated and dried extract contains less than 20 ppm, preferably less than 10 ppm and in particular less than 2 ppm 4′-O-methyl pyridoxine and/or less than 20 ppm, preferably less than 10 ppm and in particular less than 5 ppm biflavones [202,210].

Inventors from China in 2017 [211] patented a new method of obtaining flavonoids from ginkgo leaves by using fermentation with *Aspergillus niger*, which allows reducing the amount of undesired ginkgo acid in the extract. Moreover, the obtained extract is free of organic solvents. According to this procedure, cleaned and crushed leaves are mixed with water (40–60%) and sterilized. In the next step, the mixture is inoculated with *Aspergillus niger*, then enzymes are added, and the fermentation is carried out for 3–6 h at 25–30 °C. Subsequently, further protease is added, and the reaction lasts for the next 3–4 h, followed by 10 min long heat termination, centrifugation, extraction with ethanol, concentrating, and finally freeze-drying.

The raw material subjected to extraction was mainly *Ginkgo biloba* leaves, fresh or dried. They were also used to make mixtures for the production of infusions, e.g., a mixture of ginkgo leaves and other herbal materials [212,213] or beer [214].

In recent years, the subject of patents has also been products made from ginkgo nuts (also called ginkgo fruits or ginkgoes), e.g., shortbread [215], vinegar [216], beverages [217,218] or wines [219,220,221,222]. In some formulas, ginkgo powder obtained earlier from ginkgo nuts was used [223,224]. An important element was to obtain products with improved taste, devoid of bitterness. Due to the insufficient taste of the obtained products or the content of undesirable ingredients, there were searched formulas containing the addition of other valuable ingredients, e.g., wine or drink in which ginkgo fruits were used together with the other raw materials like sorghum, wheat, sugar, saffron etc. [225,226,227]. Reduction of bitterness was also caused by fermentation caused by yeast in wine or beer production, as well as *Aspergillus niger* or *Lactobacillus strains* [213,228,229,230,231].

Although the application of *Ginkgo biloba* in food was always motivated by its health-promoting properties, in recent years, this appeared to be the main reason for forming new products, and inventors evidenced their health-promoting activities [212,232,233,234].

Moreover, there have been patented many devices to facilitate the extraction, harvesting of ginkgo fruits and pre-treatment of ginkgo nuts.

## 8. Conclusions

*Ginkgo biloba* is a very popular raw material used not only in medicine but also in industrial technology. The content of ginkgolides, bilobalides, flavonoids and other bioactive ingredients contributes to its wide application. Currently, ginkgo is a herbal dietary supplement (EGb761); it is also used in complementary medicine and is an additive to cosmetics [198]. Products containing it are gaining popularity all over the world. Its primary action focuses on alleviating and/or preventing CNS dysfunction by regulating the level of cytokinins, antioxidant enzymes, kinases and receptors and modifying the activity of the PAF activating factor. Currently, more and more studies are carried out on the health-promoting properties of ginkgolic acids: their anti-cancer, neuroprotective and anti-bacterial activity is being tested. High hopes are associated with the possible medical use of ginkgolic acid. Although in 2000 years old traditional Chinese medicine, *Ginkgo biloba* seeds are used in the treatment of cough, asthma, tuberculosis, bladder infections, flatulence and diarrhea, however, such use of *Ginkgo biloba*, along with the development of knowledge about its active ingredients, brings a lot of concerns. The largest of these are the interactions between biologically active substances contained in *Ginkgo biloba* and drugs. Until now, not all the mechanisms by which the use of non-standardized extracts of *Ginkgo biloba* leaves can be used are known to cause excessive activity or inhibition of the drug’s action.

In conclusion, *Ginkgo biloba* is still an interesting research object for scientists dealing with, among others, medicine and food production. New products containing extracts or fractions of *Ginkgo biloba* fruit or leaves are being developed. So far, patented food products are not popular on the European food market, with some exceptions—products containing the addition of dried ginkgo leaves or *Ginkgo biloba* extracts, such as health beverages, are valued. Of great importance, there is the possibility of using the active substances contained in the leaves and seeds for the production of the so-called superfoods, other than dried leaf infusions. Focusing on the purification of Ginkgo biloba extracts can contribute to increasing the offer and improving the quality of dietary supplements and medicines. In addition, the development of a method for the isolation of specific *Ginkgo biloba* metabolites can provide compounds used for food enrichment.

It is worth noting that expanding the market with Ginkgo biloba products may have a positive impact on the development of Europe’s agricultural economy. This will result in no need to import the plant, e.g., from Asian countries, which seems to be a greener solution.

## Figures and Tables

**Figure 1 molecules-28-03993-f001:**
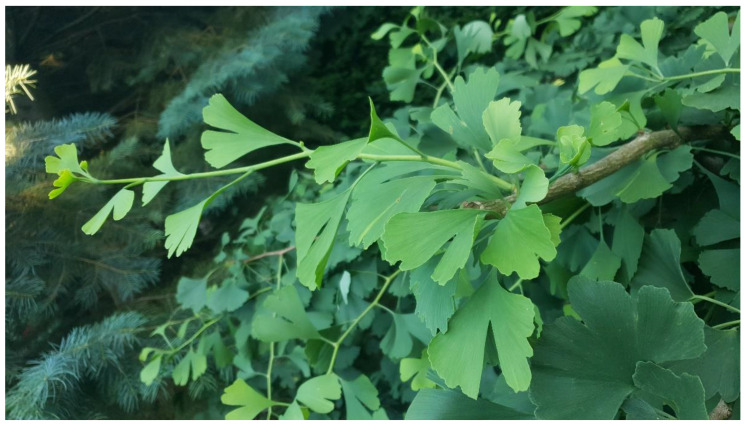
*Ginkgo biloba* stems and leaves (Rose Garden “Różanka”, Szczecin, 2021; fot. P. Biernacka).

**Figure 2 molecules-28-03993-f002:**
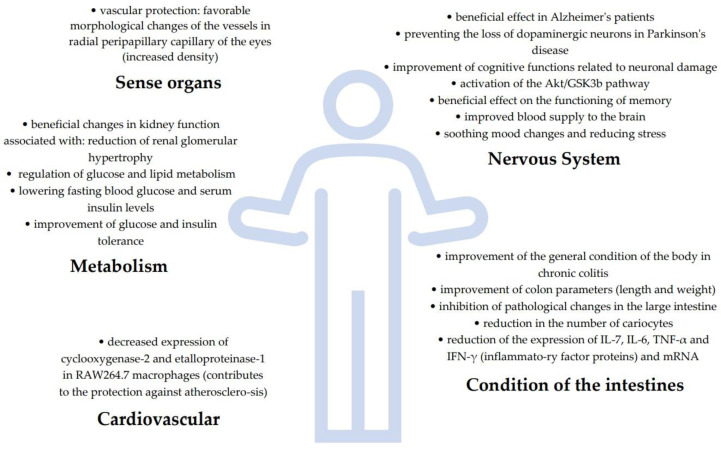
Selected health-promoting effects of Ginkgo biloba extract on the human body.

**Table 1 molecules-28-03993-t001:** Selected biologically active substances contained in *Ginkgo biloba* (leaves, fruit, roots).

Compound	Structure	Molecular Formula	Molecular Weight [g/mol]	Main Toxicological and/or Pharmacological Effects	Source of Information
	Diterpenes				
Ginkgolide A	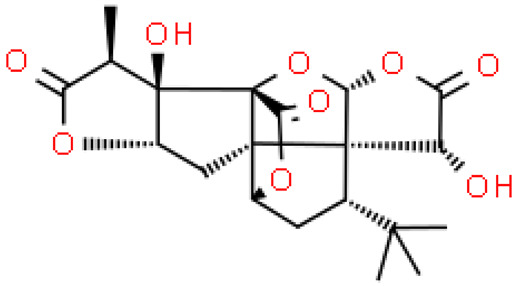	C_20_H_24_O_9_	408.399	No toxicity Anti-inflammatory and immunostimulating effect	[33,34]
Ginkgolide B	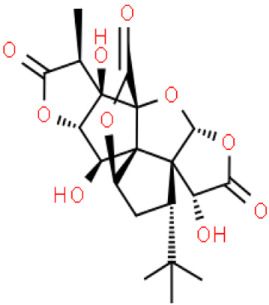	C_20_H_24_O_10_	424.399	No toxicity Beneficial effect on the functioning of the central nervous system	[33,35]
Ginkgolide C	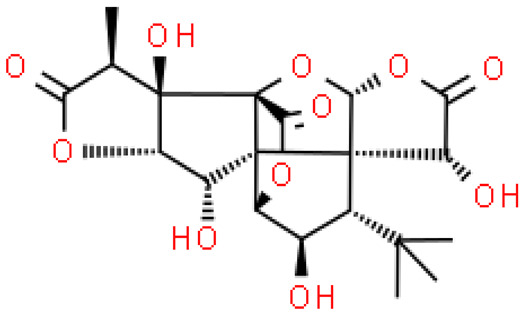	C_20_H_24_O_11_	440.398	No toxicity Reduces the accumulation of lipids, anti-cancer effect	[33,36]
Ginkgolide M	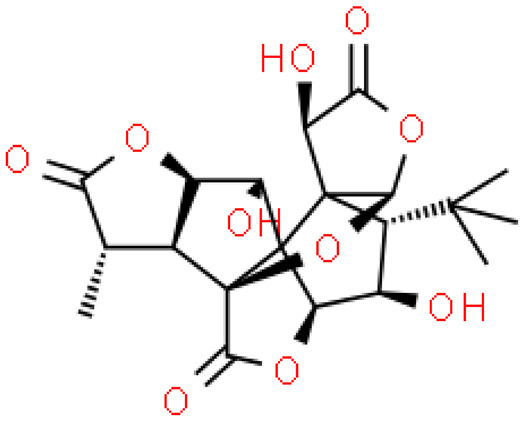	C_20_H_24_O_10_	424.399	No toxicity Inhibitor of ligand-gated ion channels in the central nervous system	[33,37]
Ginkgolide J	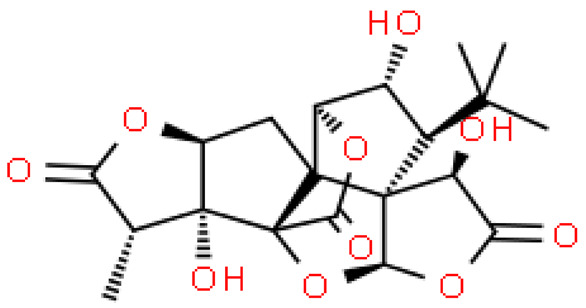	C_20_H_24_O_10_	424.399	No toxicity Dementia treatment	[33,38]
Ginkgolide P	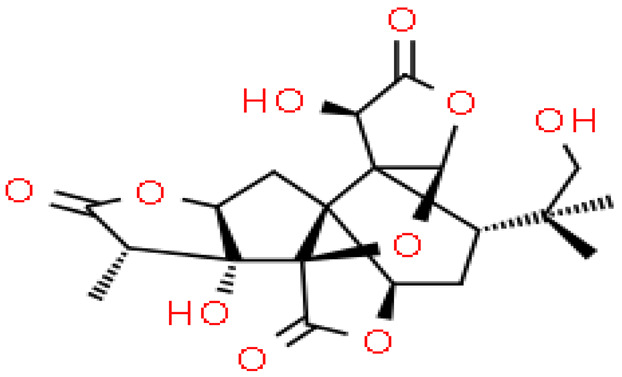	C_20_H_24_O_10_	424.399	No data	[33]
Ginkgolide Q	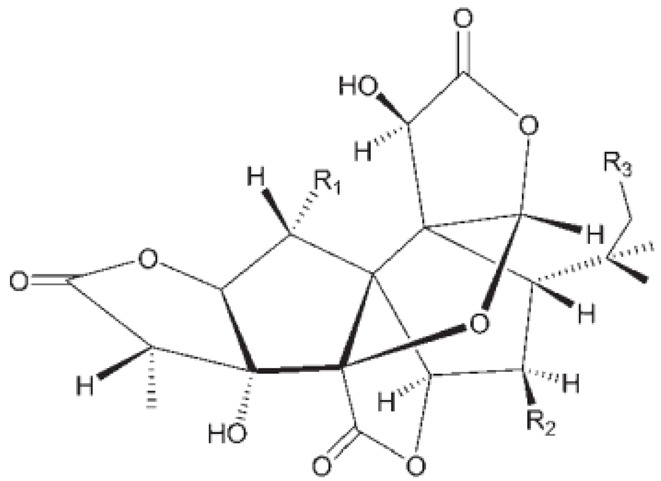	C_20_H_24_O_11_	463.126	No data	[39]
Ginkgolide K	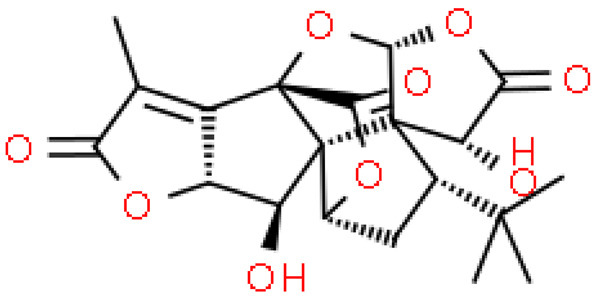	C_20_H_22_O_9_	406.383	No data Antioxidant, immunomodulatory and neuroprotective effects in ischemic stroke	[33,40,41]
Ginkgolide L	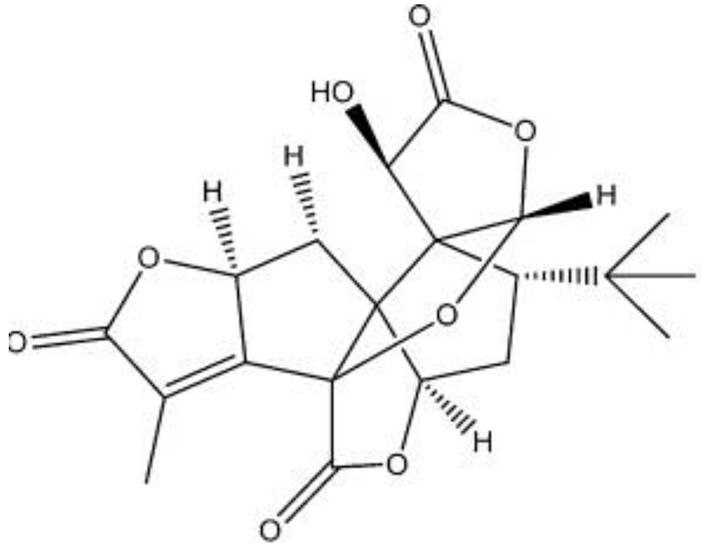	C_20_H_22_O_8_	No date	No data	[42]
Ginkgolide N	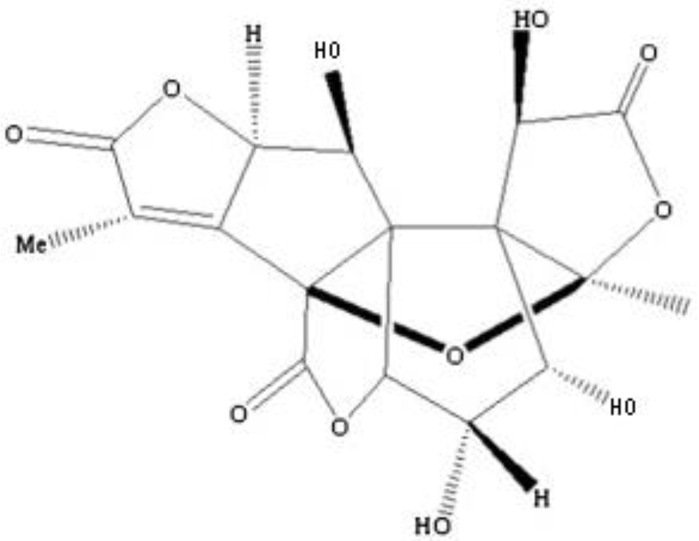	C_20_H_24_O_11_	No data	No toxicity Protective effect on damaged PC12 cells induced by glutamate	[43,44]
	Sesquiterpenes				
Bilobalide	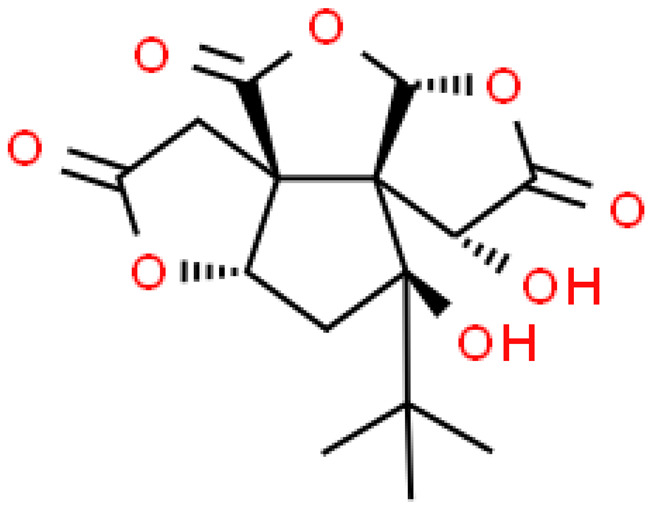	C_15_H_18_O_8_	326.299	May cause arrhythmia Neuroprotective, anti-inflammatory, antioxidant, anti-ischemic, protective effect on the circulatory system	[33,45]
Bilobalide isomer	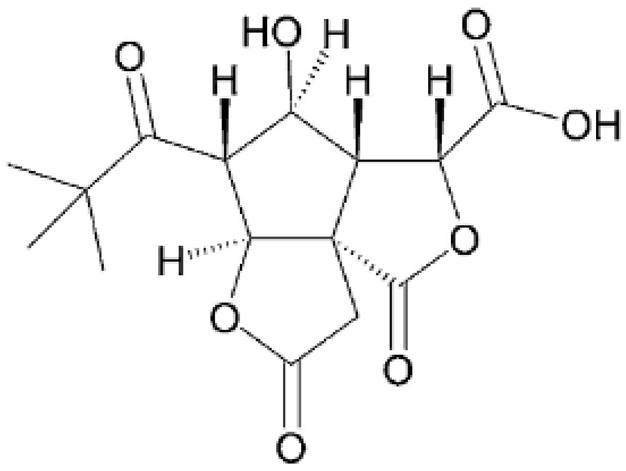	No data	No data	No data	[46]
	Flavonoids				
Quercetin	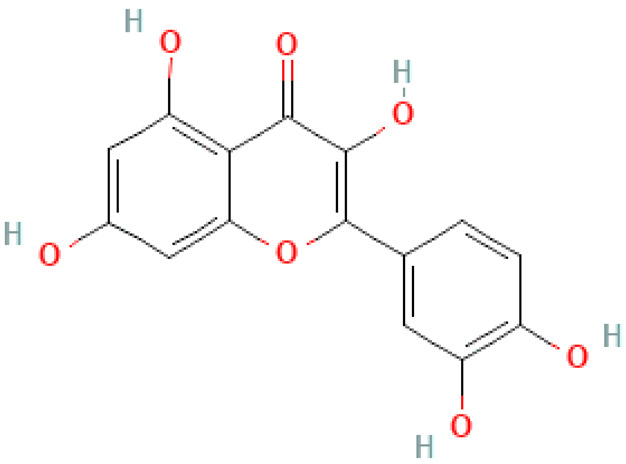	C_15_H_10_O_7_	302.23	Quercetin administration may cause cellular toxicity due to o-quinone/methide quinone side-production Anti-diabetic, anti-inflammatory, antioxidant, anti-microbial, anti-cancer effect, supporting the functioning of the circulatory and nervous systems	[10,33,47,48]
Kaempferol	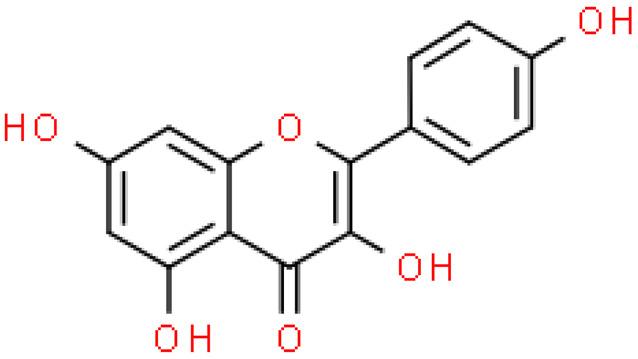	C_15_H_10_O_6_	286.236	Genotoxic and carcinogenic in vitro—no in vivo studies confirming this effect Antioxidant, anti-inflammatory, ability to scavenge free radicals	[38,49,50]
Isorhamnetin	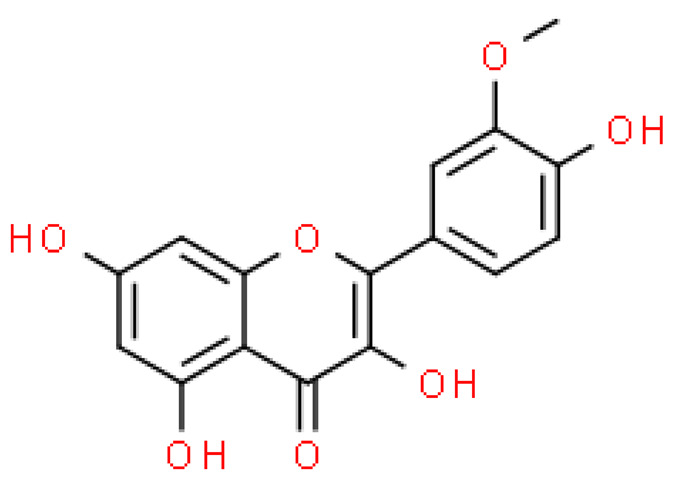	C_16_H_12_O_7_	316.262	No toxicity Protective effect on the circulatory and nervous systems, anti-atherosclerotic, hypotensive, hypoglycemic, anti-cancer, anti-inflammatory effects	[33,51]
Myricetin	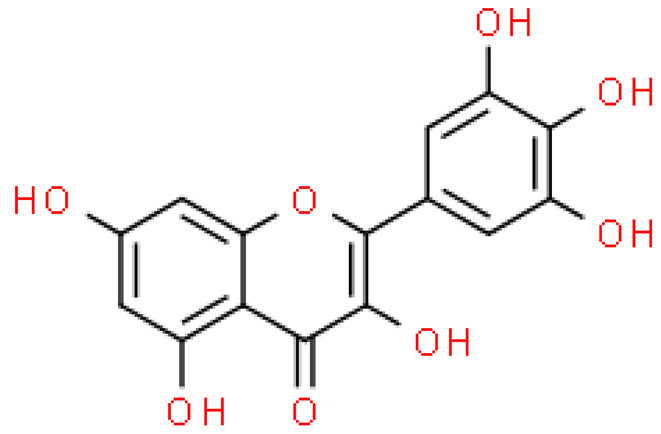	C_15_H_10_O_8_	318.235	No toxicity Antioxidant, anti-inflammatory, anti-photoaging, anti-cancer, anti-platelet aggregation, anti-hypertensive, immunostimulating effect	[33,52]
Apigenin	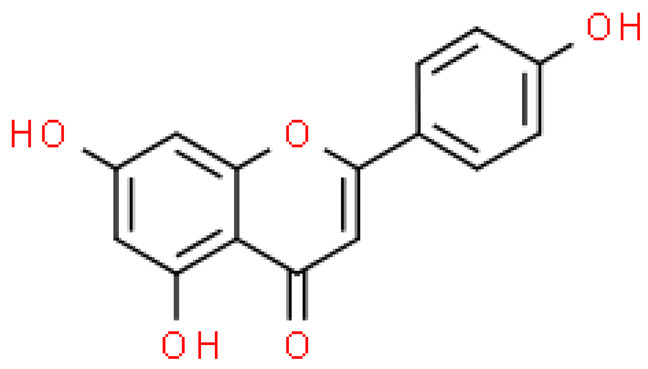	C_15_H_10_O_5_	270.237	No toxicity Anti-diabetic, anti-cancer, protective effect on the nervous system	[33,53]
Luteolin	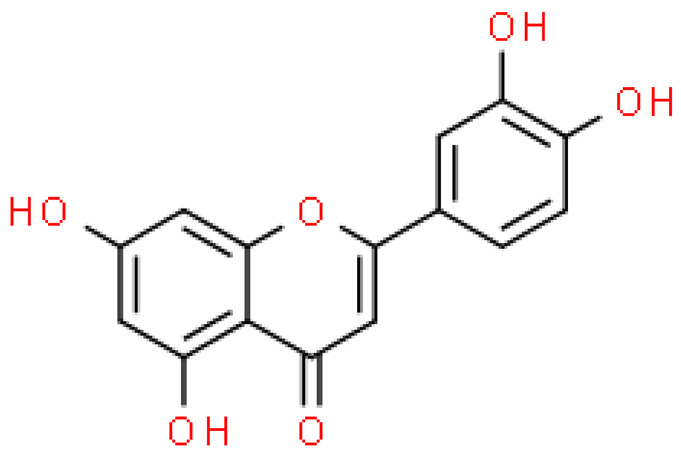	C_15_H_10_O_6_	286.236	No toxicity Antioxidant, anti-inflammatory, anti-allergic and anti-cancer effect	[33,54]
Genkwanin	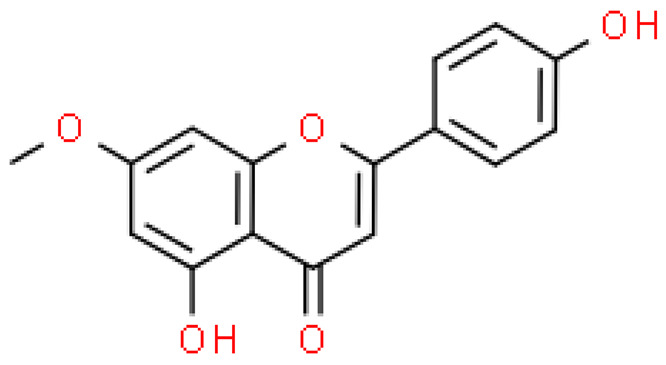	C_16_H_12_O_5_	284.263	No toxicity Anti-inflammatory, immunomodulatory, anti-bacterial, anti-rheumatic effect	[33,55]
Genistein	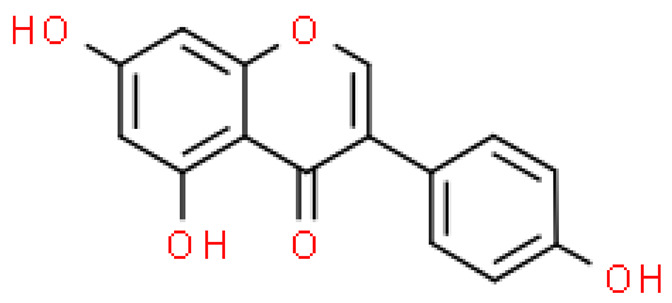	C_15_H_10_O_5_	270.237	A high dose of genistein has a strong teratogenic, endocrine-disrupting effect Anti-inflammatory effects, inhibition of nuclear factor Kappa-B, prostaglandins, pro-inflammatory cytokines, reactive oxygen species and free radical scavenging activity	[33,56]
Epicatechin	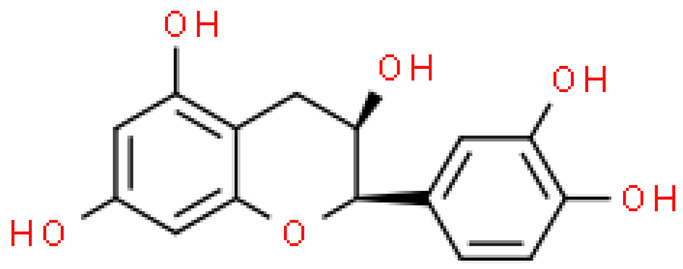	C_15_H_14_O_6_	290.268	No toxicity Antioxidant, anti-inflammatory, anti-bacterial, anti-diabetic, anti-cancer effect	[33,57]
Catechin	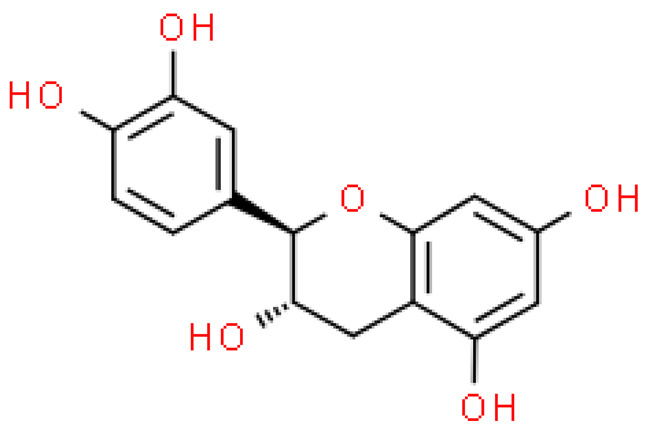	C_15_H_14_O_6_	290.268	Excessive dose may cause hepatitis Anti-cancer, anti-obesity, anti-diabetic, anti-inflammatory, anti-cardiovascular, anti-infective, hepatoprotective and neuroprotective	[33,58]
Epigallocatechin	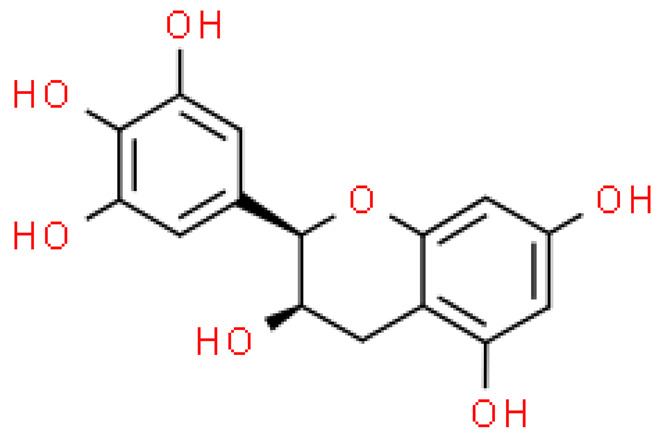	C_15_H_14_O_7_	306.267	Mild and acute health problems after using higher doses, i.e., skin irritation, hepatitis, hypoglycemia, dizziness—human and animal studies Anti-obesity, anti-microbial, anti-cancer, anti-inflammatory effect	[33,59]
Gallocatechin	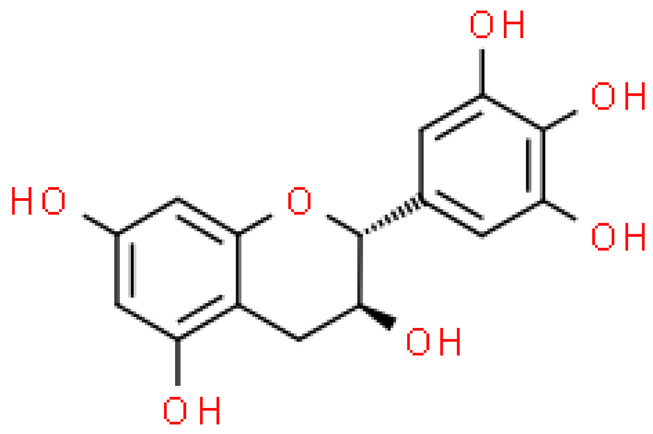	C_15_H_14_O_7_	306.267	May cause irritation of the respiratory tract (manifested by coughing and shortness of breath), skin and acute eye irritation Antioxidant and neuroprotective effect anti-diabetes, antivirus activities	[33,60,61,62,63,64]
Amentoflavone	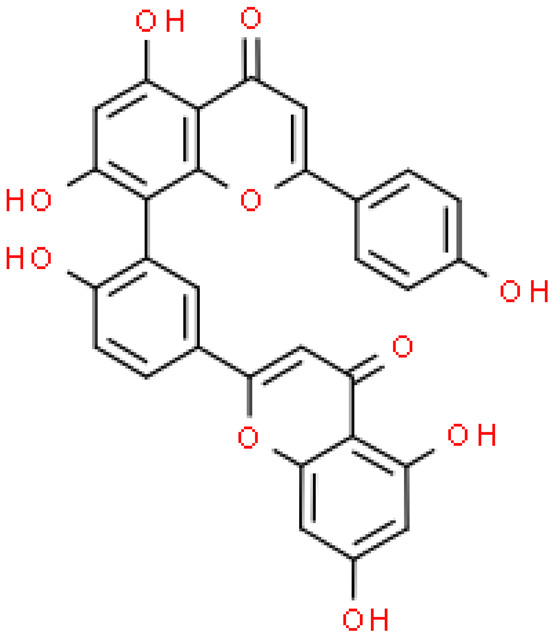	C_30_H_18_O_10_	538.458	It can be a strong inhibitor of some genes, e.g., CYP2C9 Anti-inflammatory, anti-microorganism, antioxidant, anti-angiogenesis, neuroprotective, musculoskeletal protection, radioprotection, metabolism regulation, anxiolytic/antidepressant, anti-cancer	[33,65]
Bilobetin	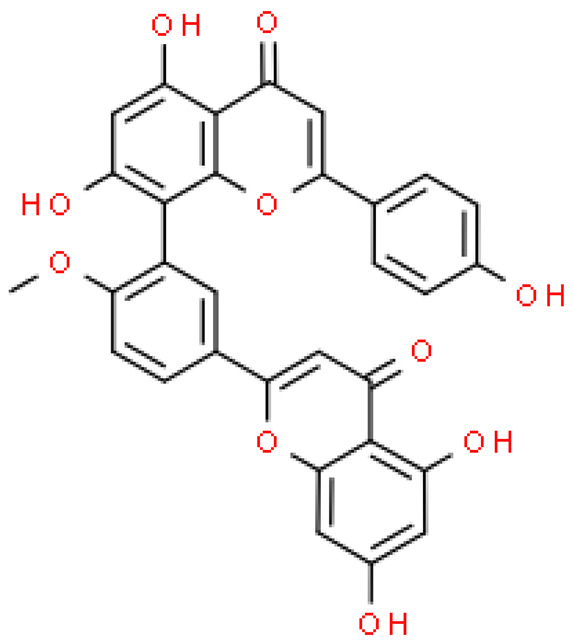	C_31_H_20_O_10_	552.484	Extensive watery degeneration of hepatocytes Antifungal, anti-inflammatory, antioxidant, antihyperlipidemic and antiproliferative effects	[33,66,67]
Sequoiaflavone	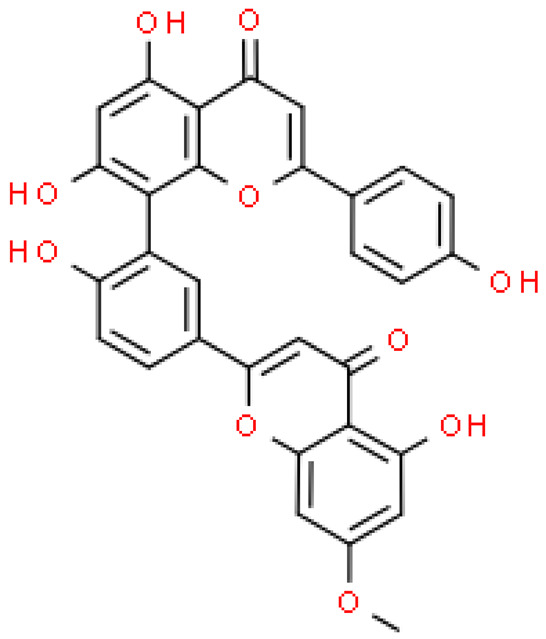	C_31_H_20_O_10_	552.484	LD toxicity in mice after oral and intraperitoneal administration at a dose above 3 gm/kg Anti-cancer activities	[33,46,64,68]
Ginkgetin	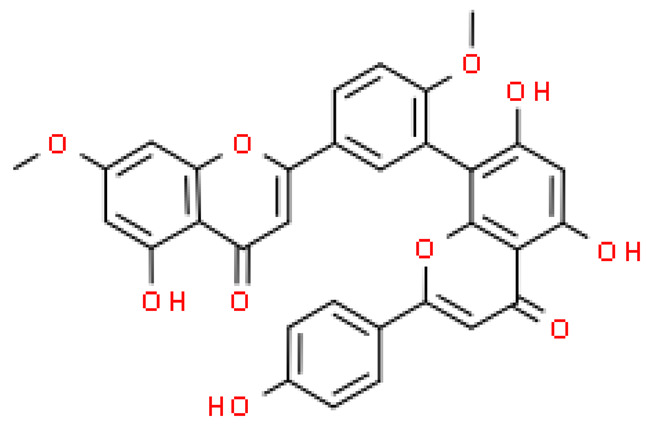	C_32_H_22_O_10_	566.511	Extensive watery degeneration of hepatocytes Anti-cancer, anti-inflammatory, anti-microbial, anti-adipogenic and neuroprotective effect	[33,66,69]
	Alkylophenolic acid				
Ginkgolic acid (C13:0)	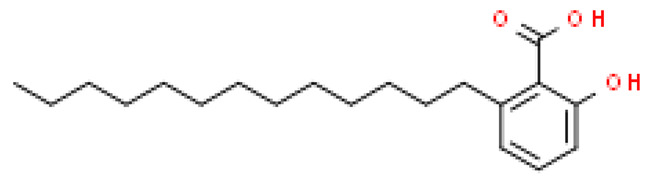	C_22_H_32_O_3_	320.466	Cytotoxic, mutagenic, genotoxic, allergenic and neurotoxic in high doses Anti-inflammatory and anti-cancer, anti-diabetic, anti-fibrotic, anti-bacterial, anti-viral and reno/neuroprotective effects	[6,33,70]
Ginkgolic acid (C15:1)	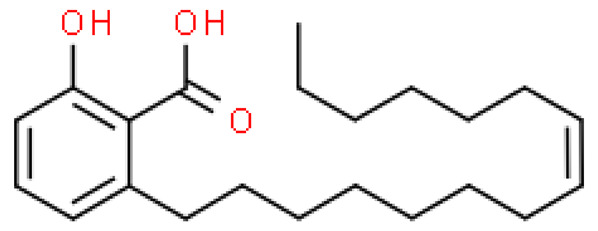	C_22_H_34_O_3_	346.504	Cytotoxic, mutagenic, genotoxic, allergenic and neurotoxic in high doses Anti-inflammatory and anti-cancer, anti-diabetic, anti-fibrotic, anti-bacterial, anti-viral and reno/neuroprotective effects	[6,33,70]
Ginkgolic acid (C17:1)	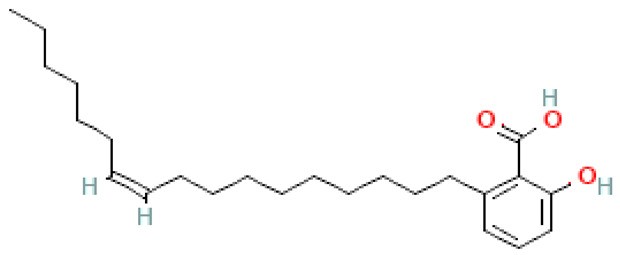	C_24_H_38_O_3_	374.600	Cytotoxic, mutagenic, genotoxic, allergenic and neurotoxic in high doses Anti-inflammatory and anti-cancer, anti-diabetic, anti-fibrotic, anti-bacterial, anti-viral and reno/neuroprotective effects	[6,33,70]
Ginkgolic acid (C17:2)	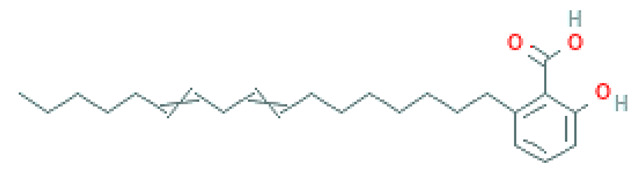	C_24_H_36_O_3_	372.500	Cytotoxic, mutagenic, genotoxic, allergenic and neurotoxic in high doses Anti-inflammatory and anti-cancer	[6,33,71]
	Alkylphenols				
Cardanols (C15:0)	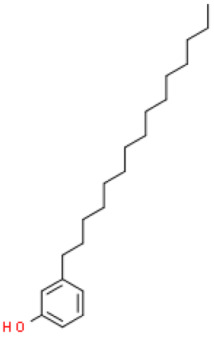	C_21_H_36_O	304.510	No data	[33]
Cardanols (C15:1)	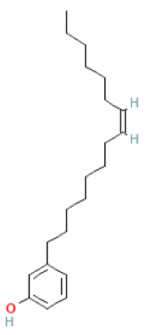	C_21_H_34_O	302.500	At high doses genotoxic effects Antioxidant, anti-cancer and antimutagenic effect. At low dose DNA damage repair	[46,72,73]
Cardol (C15:0)	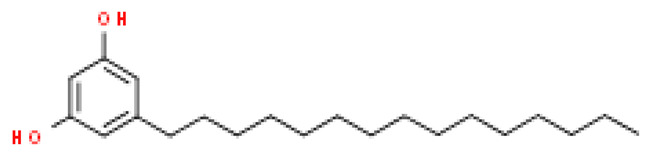	C_21_H_36_O_2_	320.509	Cytotoxic effect Anti-cancer effect—inhibits the proliferation of cancer cells and induces the death of cancer cells; antioxidant effect, neuroprotective effect	[33,74,75,76,77]
Cardol (C15:1)	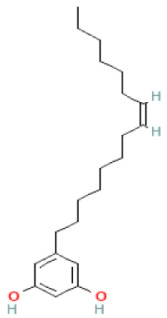	C_21_H_34_O_2_	318.5000	Cytotoxic effect No data	[46,74]
Urushiol (C15:0)	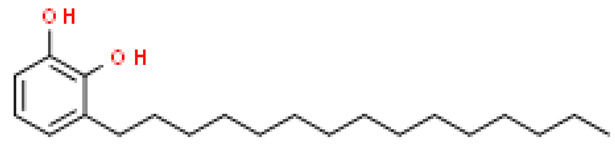	C_21_H_36_O_2_	320.509	Allergenic effect (acute inflammation of the skin) Anti-bacterial effect, anti-cancer effect (cytotoxic against tumor cells)	[33,78,79,80]
	Proanthocyanidins				
Epicatechin-(4β→8)-catechin	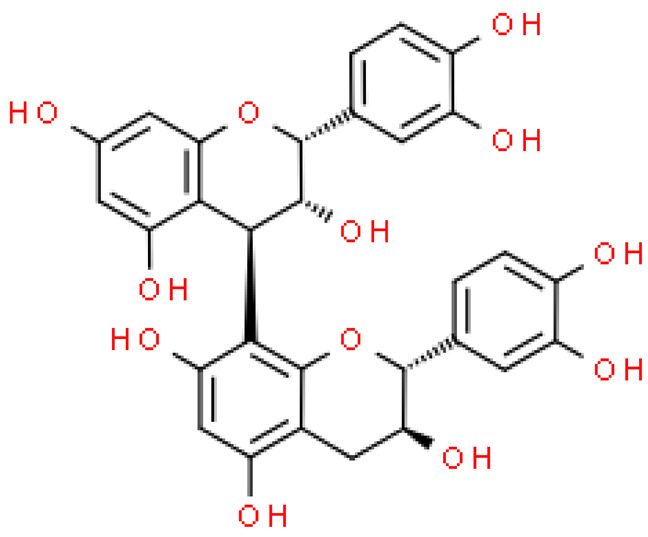	C_30_H_26_O_12_	578.520	Anti-microbial activity and strong cytotoxicity against tumor cells	[33,81]
Gallocatechin-(4β→8)-catechin	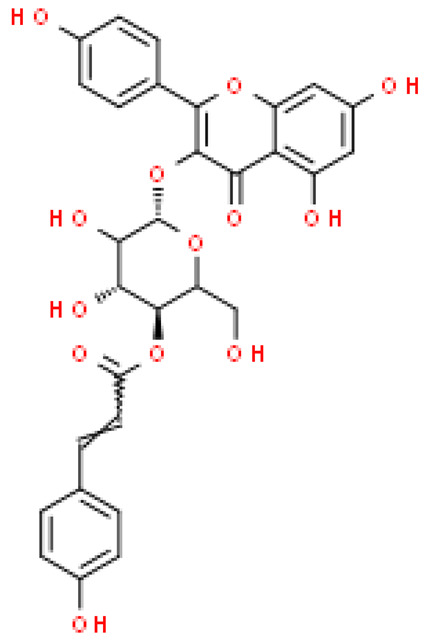	C_30_H_26_O_13_	594.520	No data	[33]
Epiallocatechin-(4β→8)-gallocatechin	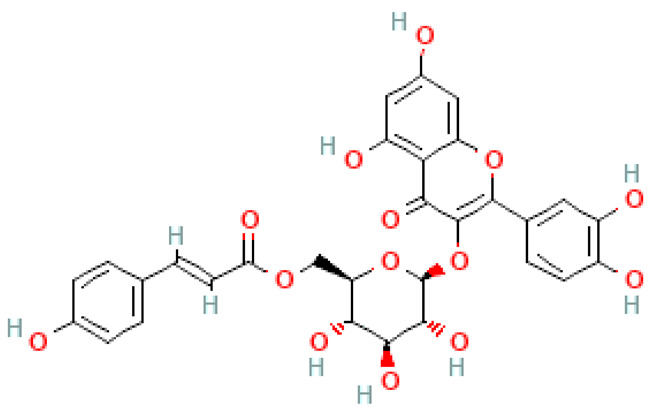	C_30_H_26_O_14_	610.500	No data Changes in fat metabolism in hyperlipidemia	[46,82]
	Carboxylic acids				
Protocatechuic acid	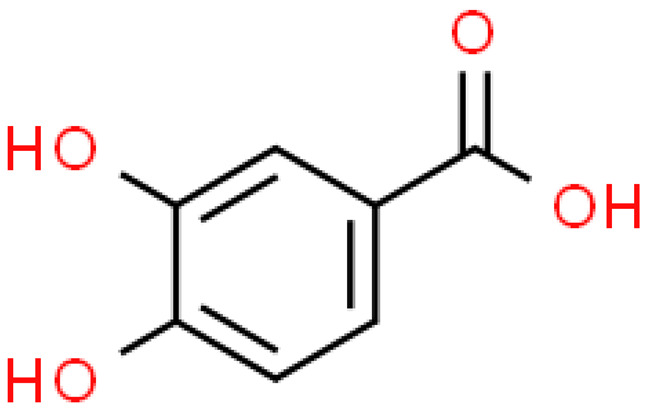	C_7_H_6_O_4_	154.120	Cytotoxic, genotoxic, carcinogenic, hepatotoxic and nephrotoxic at high doses Antioxidant, anti-inflammatory, anti-diabetic, antihypertensive, anti-atherosclerotic, anti-aging, anti-cancer, neuroprotective, anti-bacterial, anti-viral effect and protective effect for organs	[33,83]
p-hydroxybenzoic acid	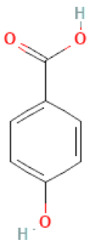	C_7_H_6_O_3_	138.120	Possible reproductive risk and potential involvement in breast cancer Antioxidant, anti-bacterial, antimutagenic, anti-thrombotic and estrogenic activity	[46,84,85,86,87,88,89]
Vanillic acid	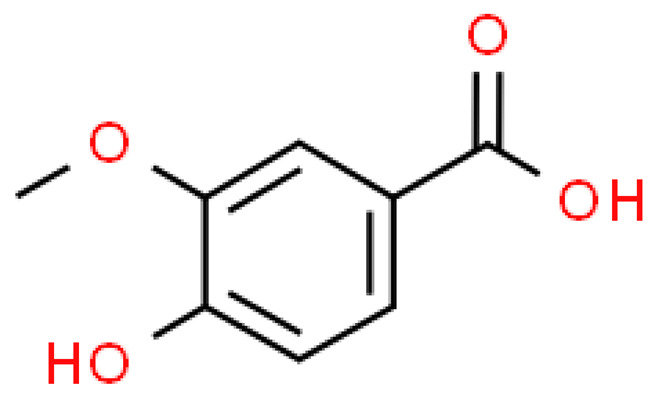	C_8_H_8_O_4_	168.147	No toxicity Sedative, anti-depressant, antioxidant, anti-hypertensive, anti-nociceptive, anti-cancer, anti-fungal, reducing the severity of ulcerative colitis, hepatoprotective, wound healing	[33,90,91]
Isovanillic acid	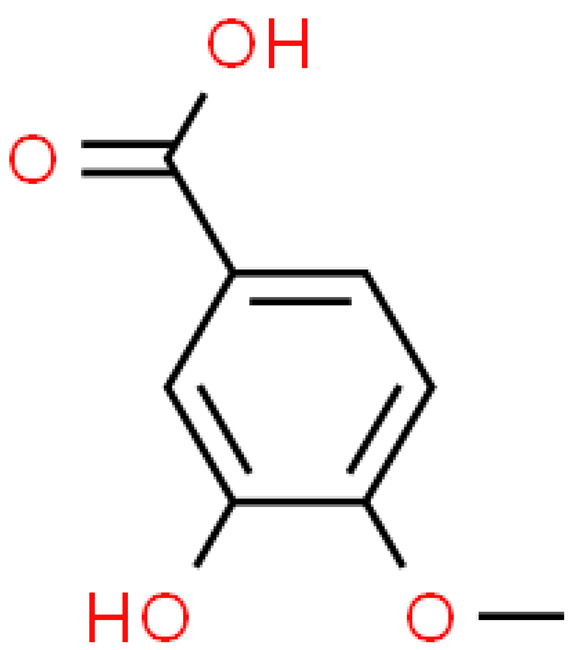	C_8_H_8_O	168.147	No toxicity Anti-thrombotic and cytostatic activity	[33,89,92,93]
Gallic acid	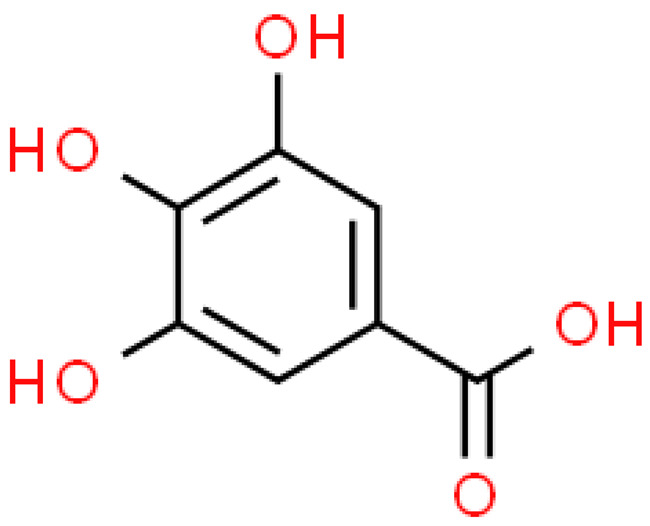	C_7_H_6_O_5_	170.120	At higher concentrations it can be toxic, e.g., cytotoxic effect. In vivo studies, the toxicity is relatively low Anti-inflammatory, antioxidant, anti-cancer, anti-bacterial, anti-diabetic, anti-obesity, anti-microbial, anti-myocardial ischemia	[33,94]
p-coumaric acid	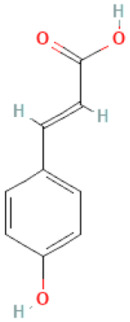	C_9_H_8_O_3_	164.160	No toxicity Anti-mutagenic, anti-genotoxic, antioxidant, anti-microbial activity, inhibits cellular melanogenesis and plays a role in immune regulation in humans	[46,95]
Caffeic acid	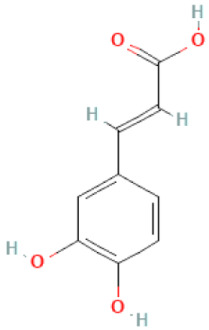	C_9_H_8_O_4_	180.160	It is anti-implantation during early pregnancy in mice at high doses Anti-inflammatory, antioxidant, anti-cancer, immunomodulatory and neuroprotective effect	[46,96,97]
Sinapic acid	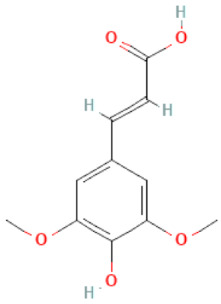	C_11_H_12_O_5_	224.21	May be cytotoxic at high doses Antioxidant, anti-inflammatory, anti-cancer, anti-hyperglycemic, anti-diabetic, anti-hypertensive, hepatoprotective, renoprotection, neuroprotective, anxiolytic, anti-bacterial effect	[46,98]
Ferulic acid	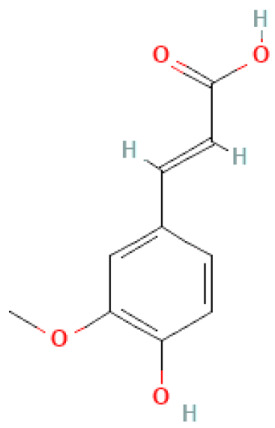	C_10_H_10_O_4_	194.180	Weak toxicity, e.g., on platelets, white and red blood cells Antioxidant, anti-inflammatory, anti-fibrotic, anti-apoptotic, anti-platelet, anti-bacterial, protective effect on vascular endothelial cells	[46,99]
Chlorogenic acid	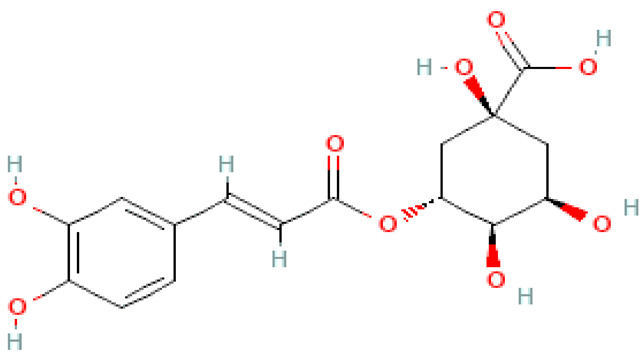	C_16_H_18_O_9_	354.310	No toxicity Neuroprotective, anti-cancer, anti-bacterial, protective effect on the circulatory system, renoprotection, protective effect on the digestive system, hepatoprotection, support in the treatment of metabolic syndrome	[46,100]
Ascorbic acid	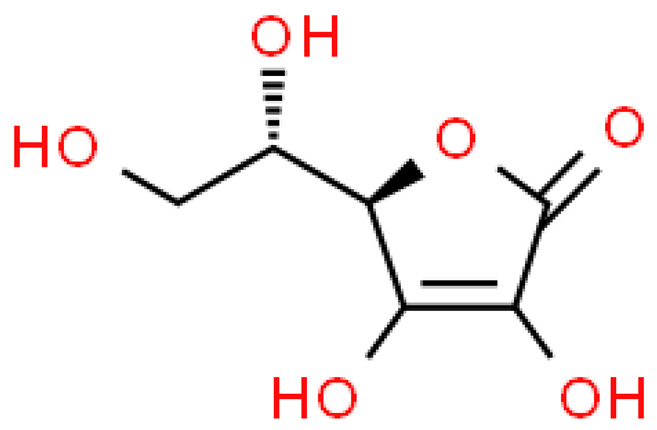	C_6_H_8_O_6_	176.124	No toxicity It has an antioxidant effect, stimulates the production and activation of immune cells	[33,101]
Quinic acid	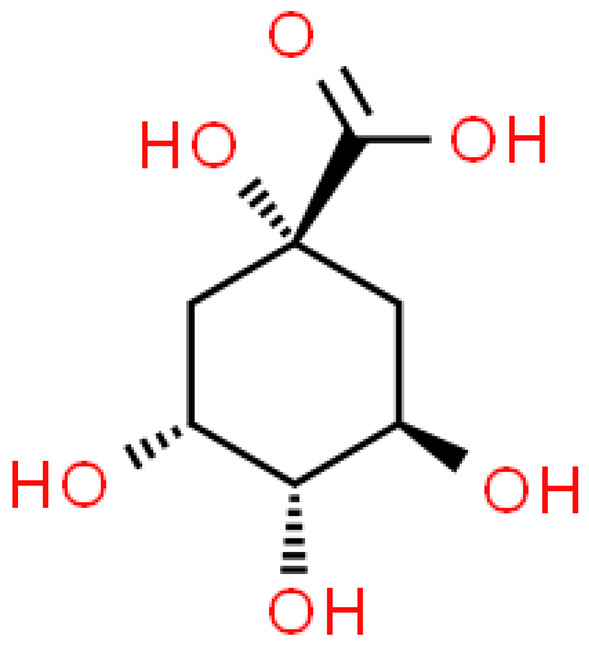	C_7_H_12_O_6_	192.167	No toxicity Antioxidant, anti-diabetic, anti-cancer, anti-microbial, anti-viral, anti-aging, protective and analgesic effects	[33,102]
Shikimic acid	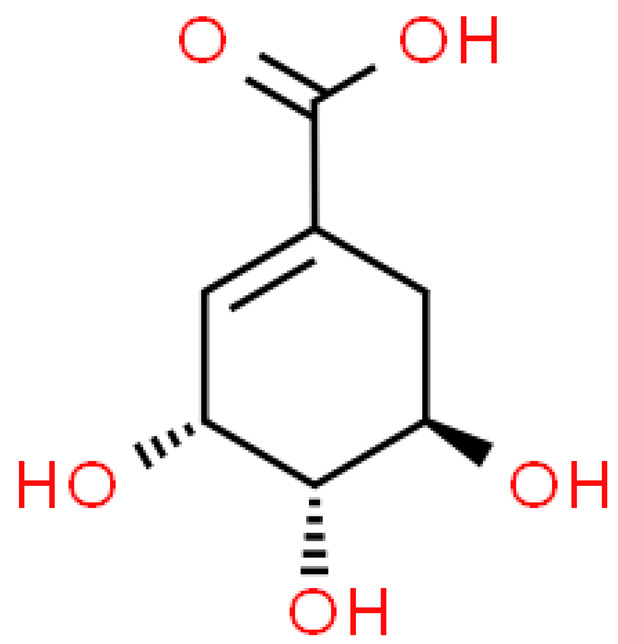	C_7_H_10_O_5_	174.151	No toxicity Antioxidant, anti-inflammatory, anti-viral, antifungal, exfoliating, anti-acne, whitening, moisturizing, anti-aging, sebum-regulating, hair growth stimulating	[33,103]
	Lignans				
Sesamin	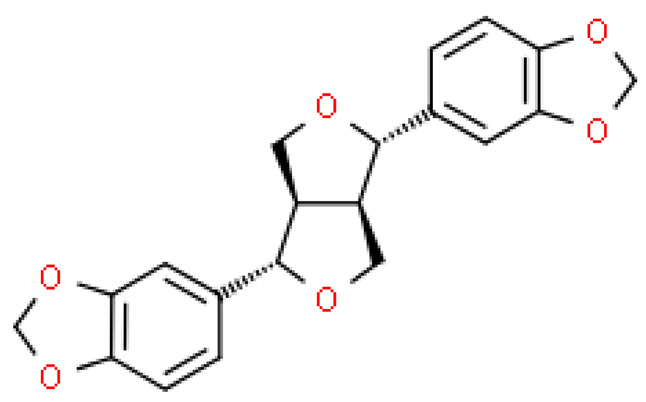	C_20_H_18_O	354.353	In high doses, it can be a compound with a low and moderate degree of danger, e.g., it can cause loss of appetite, vomiting, diarrhea, hormone metabolism disorders Antioxidant and anti-inflammatory, anti-hypertensive, anti-atherosclerotic, lipolytic, anti-thrombotic, anti-diabetic and anti-obesity effects	[33,104]
Ginkgool	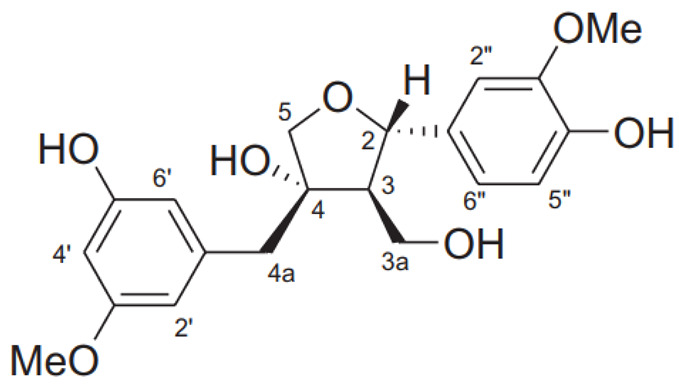	C_20_H_24_O_7_	376.000	No data	[105]
Pinoresinol	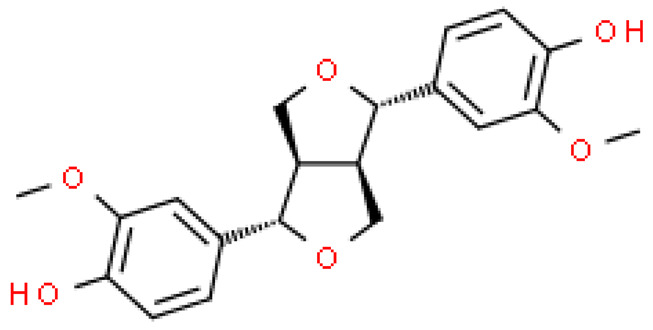	C_20_H_22_O_6_	358.385	No toxicity Hypoglycemic effect, improving memory and learning ability, anti-cancer effect (stimulation of cancer cell apoptosis)	[33,106,107,108,109]
Ginkgolide B	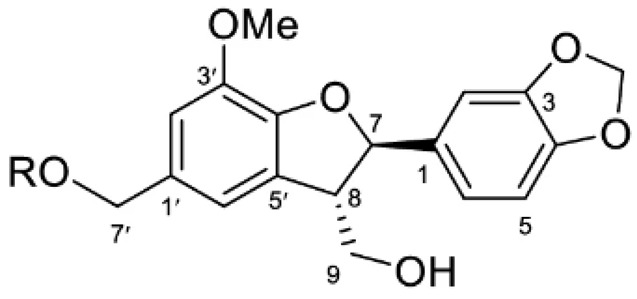	C_24_H_28_O_11_	492.000	No toxicity Anti-inflammatory and anti-aging effect	[9,110]
p-hydroxyphenyl	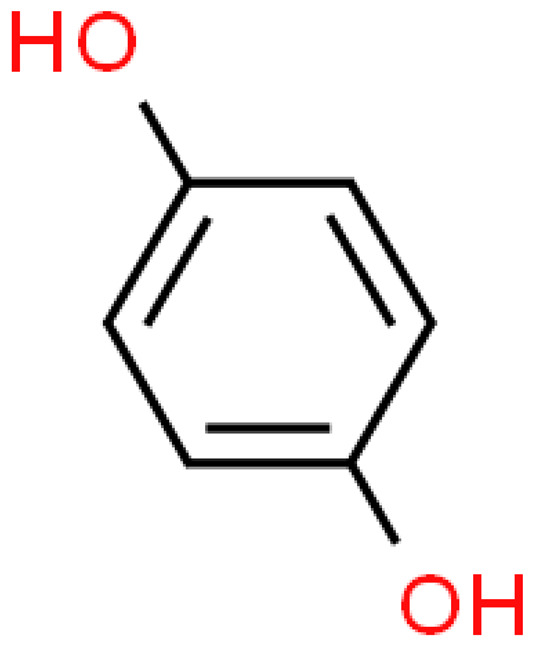	C_6_H_6_O_2_	110.111	Oral administration causes acute poisoning (abdominal pain, vomiting, tachycardia, convulsions, convulsions and coma) or formation of neoplastic lesions; skin contact may cause irritation (discoloration or erythema) and allergic dermatitis Treatment of melasma and post-inflammatory hyperpigmentation of the skin (tyrosinase inhibitor)	[33,111,112,113]

## Data Availability

Not applicable.

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
