# Peer review of "The Potential of Ginkgo biloba as a Source of Biologically Active Compounds—A Review of the Recent Literature and Patents"

_molecules, 2023, doi:10.3390/molecules28103993_

Round 1

Reviewer 1 Report

The authors have summarized the toxicological and pharmacological activities of the metabolites in Ginkgo biloba in details. In addition, the authors have given a brief summary on the patents related to G. biloba. Overall, the review was well written and organized. The contexts were also suitable theme of the special issue.

Before accepting the manuscript, the authors should add a column next to the molecular weights of each metabolite listed in Table 1 and provide the key toxicological and/ or pharmacological effects of each metabolite in that column. This will help the audiences to know the bioactivities of each metabolite.

Author Response

I would like to thank you very much for the positive review. I followed your suggestion and added a column with major toxicological and pharmaceutical effects.

Thank you very much.

Reviewer 2 Report

The review “The potential of Ginkgo biloba as a source of biologically active compounds - a review of the recent literature and patents” represents the modern analysis and the latest research results of Ginkgo biloba with high medicinal value, so this review seems to be important and actual. The main recommendations are the following: 1) Table 1 contains the structures of the main components of Ginkgo biloba of different classes but many of them are well known and it is not necessary to demonstrate their formula. May be only new components isolated in the period from 2018-2022 should be demonstrated (but without bruto and molecular weight). The other well-known structures could be moved to Supporting file.  2) the representative Figure with all discussed activities (mechanisms of action) is necessary. 3) the authors should formulate clear perspectives in what direction do they see the scientific development of Ginkgo biloba story in the next few years.

Author Response

I would like to thank you very much for the positive review. I have applied the suggested changes. I added a drawing showing the mechanism of action and better shrunk the prospects of using Ginkgo biloba in the future.

Due to the suggestion of the second reviewer regarding the extension of the table with toxicological and pharmacological effects of individual compounds, I decided to extend the table, which significantly emphasized its importance.

Thank you very much.